# Toward Artificial Palpation: Representation Learning of Touch on Soft Bodies

**Zohar Rimon**[*]  **Elisei Shafer**  **Tal Tepper**  **Efrat Shimron**  **Aviv Tamar**

Technion - Israel Institute of Technology

## Abstract

Palpation, the use of touch in medical examination, is almost exclusively performed by humans. We investigate a proof of concept for an artificial palpation method based on self-supervised learning. Our key idea is that an encoder-decoder framework can learn a *representation* from a sequence of tactile measurements that contains all the relevant information about the palpated object. We conjecture that such a representation can be used for downstream tasks such as tactile imaging and change detection. With enough training data, it should capture intricate patterns in the tactile measurements that go beyond a simple map of forces – the current state of the art. To validate our approach, we both develop a simulation environment and collect a real-world dataset of soft objects and corresponding ground truth images obtained by magnetic resonance imaging (MRI). We collect palpation sequences using a robot equipped with a tactile sensor, and train a model that predicts sensory readings at different positions on the object. We investigate the representation learned in this process, and demonstrate its use in imaging and change detection.

## 1 Introduction

Palpation, the use of touch in medical examination, is a centuries-old practice that is still important today. In woman's breast cancer – a case that motivates our work, a large fraction of cases (over 40% in the US, per Roth et al. 2011) are discovered by palpation, either via self-examination or by a physician, although screening tests based on mammography (X-ray) and magnetic resonance imaging (MRI) are designed to detect tumors that are not yet palpable. The recommendations of the American Cancer Society [2023] for women to be familiar with how their breasts normally feel and report any changes promptly indicate that tactile information is still relevant for early detection.

Artificial palpation has the potential to better exploit tactile information by detecting patient-specific temporal changes that physicians typically cannot keep track of, and by improving palpation precision beyond what untrained patients can achieve. Tactile imaging methods [Sarvazyan et al., 2012] typically involve pressing a force sensor array against soft tissue to generate a force map, which can then be analyzed either visually or via computer vision algorithms. In contrast, elastography [Sarvazyan et al., 2011] infers the elastic properties and stiffness of soft tissue by measuring variations in ultrasound (US) or magnetic resonance (MR) signals in response to applied forces. Additionally, several studies have investigated the use of tactile sensors for classifying tissue based on stiffness [Jia et al., 2013, Nichols and Okamura, 2015, Di et al., 2024].

Aimed at improving tactile imaging and detection accuracy, we propose to go beyond direct stiffness estimation and view palpation as an *inference process* of mechanical structures in a soft body, given a sequence of partial, noisy, tactile force measurements. Indeed, a physician performing a breast examination tries to infer the existence of lumps, cysts, and other anatomical structures from touch. The medical literature that characterizes, for example, benign masses as 'smooth, soft to firm, and

---

[*]Correspondence to zohar.rimon@campus.technion.ac.il. Code and data are available at github.com/zoharri/ArtificialPalpation

mobile, with well-defined margins' [Klein, 2005], hints that human palpation relies on more involved characterization of how different structures give in to finger *motions* rather than stiffness alone.

While palpation can be taught and perfected, human touch is a *general* skill, learned from years of physical interactions and manipulations of objects. We argue that a data-driven approach to artificial palpation can learn, from touching many different objects, how to interpret a sequence of tactile measurements into corresponding mechanical structures, potentially leading to more accurate imaging than currently available.

Motivated by self-supervised learning results in computer vision and natural language processing [He et al., 2022, Devlin et al., 2019], we propose to learn a general, artificial, palpation *representation*, by predicting future tactile force measurements given a sequence of past measurements. If the representation is useful for predicting future forces, it must contain relevant information about the object being palpated, and may therefore be useful for tactile imaging, change detection, and classification of suspicious findings.

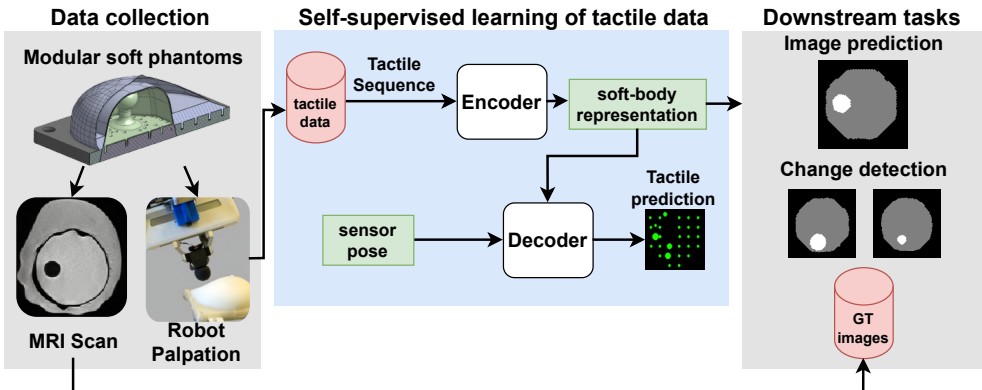

Figure 1: Proof of concept system for learning artificial breast palpation. Left: we fabricate soft objects and palpate them using a tactile sensor mounted on a robot arm. We also obtain MRI scans of the objects as ground truth object models. Middle: we train an encoder-decoder neural network to predict the tactile measurements at given positions from a sequence of previous measurements. Right: we use the learned representation to train a model for tactile imaging, and perform change detection based on predicted images. In principle, by replacing the phantoms with human subjects, our system can be used for clinical studies.

We present a proof of concept system that includes object fabrication techniques, data collection routines, model training, and an evaluation protocol of tactile imaging and change detection for soft bodies. Our fabricated objects are novel soft breast phantoms with a modular component that can include lumps with various sizes and shapes, resulting in over 1150 possible object configurations. We collect data using a robotic manipulator with a tactile sensor tip programmed to palpate the phantom, and train a neural representation by learning to minimize tactile force prediction error. To evaluate the utility of the representation for tactile imaging, we scan the phantoms in an MRI to generate ground truth object models, and train an additional neural network to predict this model from the palpation representation. To evaluate change detection, we use the predicted models to evaluate change in the size of the lump, and compare with human evaluations. We find that our learned representation contains relevant information about the position and shape of the lump, and yields tactile images that are arguably easier to interpret than a map of forces, which can be used for change detection at a level comparable to humans.

While clearly only a prototype, we argue that these results provide a promising direction for an artificial palpation system that uses human MRI scans and tactile data to *learn* tactile imaging.

## 2 Related Work

**Learning tactile representations:** work in this area focused almost exclusively on rigid bodies, and applications to object manipulation and shape reconstruction. Guzey et al. [2023] considered

instantaneous tactile measurements, and applied BYOL [Grill et al., 2020] to multiple taxel readings arranged as an image. Higuera et al. [2024] considered vision-based tactile sensors [Lambeta et al., 2020], proposed several self-supervised learning representations, and a benchmark for evaluating them in various robotic manipulating tasks. More recently, Feng et al. [2025] collected matched tactile data for four different sensors, and proposed an encoder-decoder framework for images and short (3-frame) videos from vision-based tactile sensors. Qi et al. [2023] represented a sequence of tactile, proprioceptive, and visual observations of a rigid body, by learning to reconstruct simulated 3D point-clouds of the object. The representation was further used for learning in-hand object rotation and shape reconstruction. Focusing on rigid-object shape reconstruction and pose estimation, Zhao et al. [2023] estimate pose and shape with neural networks, from data of manually pressing a rigid object with a visuo-tactile sensor, while Suresh et al. [2024] used neural fields to learn the pose and 3D shape of an object rotated by a robotic hand. Neither of these methods are applicable to estimating the properties of soft objects.

**Robotic Palpation and Cancer Diagnosis** Khanna and Shah [2024] provide a recent survey in the context of breast cancer diagnosis; we highlight several studies most relevant to our work. Jenkinson et al. [2023b] develop a radial robotic mechanism for breast palpation, while Syrymova et al. [2025] investigate a purpose-built tactile glove. Scimeca et al. [2022] palpate a soft silicone object with hard inclusions of 3 different sizes, and classify the size of the inclusion by projecting instantaneous measurements to their first principal component, and fitting a Gaussian density to the projected measurements over time per inclusion size. Our work is similar in spirit but significantly larger in scale, both in the data and the learning models. Di et al. [2024] used data from a vision-based tactile sensor to classify both prostate phantoms and real prostate tissue ex-vivo according their hardness, by fine-tuning a video masked auto-encoder applied to sequences of images.

Several studies explored how to define the robot's *motion* during palpation. Scimeca et al. [2022] investigated various motions that revolve around a point in space. Sanni et al. [2022] used deep movement primitives for robotic breast palpation, while Zhao et al. [2025] explored a shared autonomy scheme between a human tele-operator and computerized control. In this work we focus on simple linear motions and leave the investigation of more complex movements to future work.

In terms of phantom fabrication, most of the literature concerns fabrications suitable for MRI and ultrasound imaging [e.g., Ustbas et al., 2018, Keenan et al., 2016]. We are not aware of standard methods for fabricating phantoms for breast palpation.

## 3 Method

We propose a data-driven approach for soft-body tactile palpation. We first formulate our inference problem. We then use self-supervised learning approaches to learn a representation of the palpated object (Section 3.2). Finally, we use this representation to learn downstream tasks with limited amount of supervised data (Section 3.3).

### 3.1 Formulation

We model the artificial palpation problem as follows. A tactile sensor is a rigid body that is controlled to be at time $t \in 0, \ldots, T$, in pose $x_t \in \mathbb{R}^6$.[2] The sensor interacts with a soft body $M$, resulting in a $k$-dimensional force reading $f_t \in \mathbb{R}^k$. We consider $M$ to represent all the structural and mechanical properties of the body, which determine the force on the sensor, and note that in all practical situations $M$ is unknown. Subse-

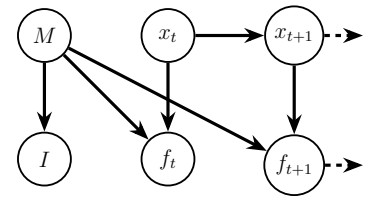

quently, the sensor is moved to the next pose $x_{t+1}$ by the controller, and the next reading is obtained. In addition, we assume access to some observation of the soft body, denoted $I$, for example, an MRI scan. In this work we do not consider *how* to control the sensor, and assume that some palpation motion controller is available. Also, we assume that the force sensor is noisy, but we do not model the noise in any explicit form.

In the `tactile imaging` problem, our goal is to use the poses and force readings $x_0, f_0, \ldots, x_T, f_T$ to predict the observation $I$. In the `change detection` problem, we are given readings from two bodies $M, M'$, which may or may not be different, and our goal is to determine if $M = M'$.

---

[2]For simplicity we consider a single sensor; our formulation trivially extends to multiple sensors.

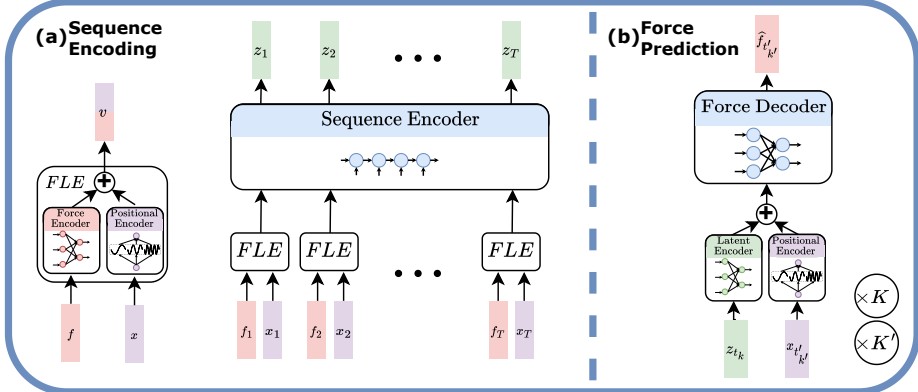

Figure 2: **Representation Learning:** (a) A sequence of tactile measurements and poses is encoded by first encoding every measurement+pose by a force-location encoder (FLE), and then encoding the sequence by a GRU. (b) The decoder predicts a tactile measurement at time $t'_{k'}$ from the representation at time $t_k$ and the pose at time $t'_{k'}$.

## 3.2 Representation Learning

To address the inference problems described above, we propose a learning based approach. We first explain the data structure, and then our learning algorithm.

We collect data from $N$ bodies $M_1, \ldots, M_N$. For each body $M_i$, we collect tactile data $\mathcal{D}_i \equiv \left\{ x^i_t, f^i_t \right\}_{t=1}^T$, by a palpation controller as described above. In addition, for the first $N_I$ bodies we have a corresponding observation $I_i$, $i \leq N_I$.

Our working hypothesis is that collecting large amounts of observations (e.g., MRI scans) is more difficult than collecting palpation data, that is, $N_I < N$. Following prior works [Devlin et al., 2019, He et al., 2022], we design our approach to first use self-supervised learning techniques to pretrain a representation from the large amount of palpation data, and later use the pretrained representation to effectively learn downstream tasks such as `tactile imaging`.

We use an encoder-decoder architecture to learn a representation in a self-supervised manner. The key idea is that the encoder is trained to map a sequence of $t$ measurements $\left\{ x^i_t, f^i_t \right\}_{t=1}^t$ from a body $M_i$ to a $d_z$-dimensional vector representation $z_t \in \mathbb{R}^{d_z}$ that contains the information in the measurements about $M_i$. Thus, at the last time step $T$, the representation $z_T$ contains information about the complete trajectory, and can be used for downstream tasks. We next detail our architecture and training objective.

**Encoder** Our encoder maps between a sequence of measurements to a sequence of latent representations. We first use a force-location encoder (FLE) to encode each step separately: $FLE(f_t, x_t) = MLP(f_t) + PE(x_t), \qquad \forall t \in [1, T]$. The forces are encoded via a two-layer MLP, and the locations are encoded with a sinusoidal positional encoding (PE). Afterwards, we use a sequence encoder to produce a sequence of embeddings of the same length as the input sequence $\left\{ z_t \right\}_{t=1}^T$. We choose a gated recurrent unit (GRU) [Cho et al., 2014] encoder as it can easily scale to long sequences, and acts as an information bottleneck in the encoding process.[3]

**Decoder** We hypothesize that if the representation is informative for predicting force measurements, it should contain relevant information about the body being palpated. Therefore, we structure our force decoder (FD) to predict the force reading at pose $x_{t'}$ based on the representation $z_t$: $FD(z_t, x_{t'}) = MLP(MLP(z_t) + PE(x_{t'})) \in \mathbb{R}^k, \qquad \forall t, t' \in [1, T]$, For $t' > t$, this corresponds to predicting future sensor readings, while for $t' < t$ this means recalling past measurements. We note that a similar idea of predicting both past and future measurements was proposed by Zintgraf et al. [2021] for decision making in partially observed domains.

---

[3]In simulation experiments that are not reported here, we also investigated other architectures such as transformer-based masked autoencoders [He et al., 2022], but obtained similar results.

**Training Objective**    We use the mean squared error (MSE) reconstruction loss between the predicted and true forces to train our encoder-decoder model. Since predicting all the forces from all the timestamps has an $\mathcal{O}\left(T^2\right)$ complexity, applying it to long sequences is challenging. Instead, we uniformly subsample reconstruction steps for calculating the loss, as described next for data from a single body $M$:

$$\mathcal{L}_{rec} = \frac{1}{2KK'} \sum_{k=1}^{K} \sum_{k'=1}^{K'} \left\| FD\left(z_{t_k}, x_{t'_{k'}}\right) - f_{t'_{k'}} \right\|^2,$$

where $\{t_k\}_{k=1}^{K}$ are $K < T$ uniform samples without replacement from $[1, T]$, and for each $k \in [1, K]$, $\{t'_{k'}\}_{k'=1}^{K'}$ are also $K' < T$ uniform samples without replacement from $[1, T]$. After calculating all representations $z_1, \ldots, z_T$ during the encoder's forward pass, a decoder forward pass and loss calculation can be done in parallel for all indices $k, k'$ and has a complexity of only $\mathcal{O}\left(K' \cdot K'\right)$. In our implementation we set $K = K' = 64$.

### 3.3    Tactile Imaging

We hypothesize that the representations $z$ learned as described in Section 3.2 are useful for predicting more general properties of the body $M$ than forces. In particular, we focus on the `tactile imaging` problem, and propose to predict the observation $I$ from the representation of a complete palpation sequence $z_T$.

We use flow matching Lipman et al. [2024], conditioned on the vector representation, to map $z_T$ to a $128 \times 128$ image where each pixel can take one of 3 different values (to be described later). During inference, we draw a single noise sample to generate an image, as the uncertainty at time step $T$ is very small (the standard deviation over results from different noise samples is two orders of magnitude smaller than the expectation), full technical details are in Section E. This architecture can easily be modified to handle more general images, or even 3-dimensional volumetric images. We train the imaging network separately from the pretrained tactile representation, using a standard cross-entropy loss for each pixel in the image.

For change detection based on the predicted images, we consider two different methods. The first compares between the pixel values of images obtained from palpation trajectories of two bodies directly. This method, which is reported in the supplementary material, works well when the image predictions are relatively accurate, as we obtained in our simulation results. For our real world results, however, we first evaluated the lump size from the predicted images, and detected change based on the predicted lump size (see Section 4).

## 4    Results

We next present our experimental results. We begin with introducing a new simulated environment for palpation, in which we investigate our design choices and quantify the scale of data required to obtain meaningful results. We then detail our real-world data collection and experiments. We provide supplementary visualizations in `zoharri.github.io/artificial-palpation`, and our code can be found in the supplementary material.

### 4.1    The PalpationSim Simulator

We introduce `PalpationSim` – a simulation environment to be used as a mock of the real palpation learning setup. We designed `PalpationSim` to be both lightweight, quick, and easy to visualize and interpret, yet indicative of our real-world domain.

Our main component is a 2-dimensional finite element method (FEM) simulation of a soft semi-circular body with the option of having a harder lump element inside, as depicted in Figure 3a.[4] The body is composed of triangular elements with linear elasticity, and we model the different hardness

---

[4]We release `PalpationSim` as part of our open-source code. While several physical simulators that are popular in the robotics literature can simulate soft objects [Todorov et al., 2012, The Isaac Lab Project Developers, 2025], we did not find any versatile enough for our simulation environment at the time of writing. On the other hand, we desired a much faster and lightweight solution than full blown commercial FEM packages.

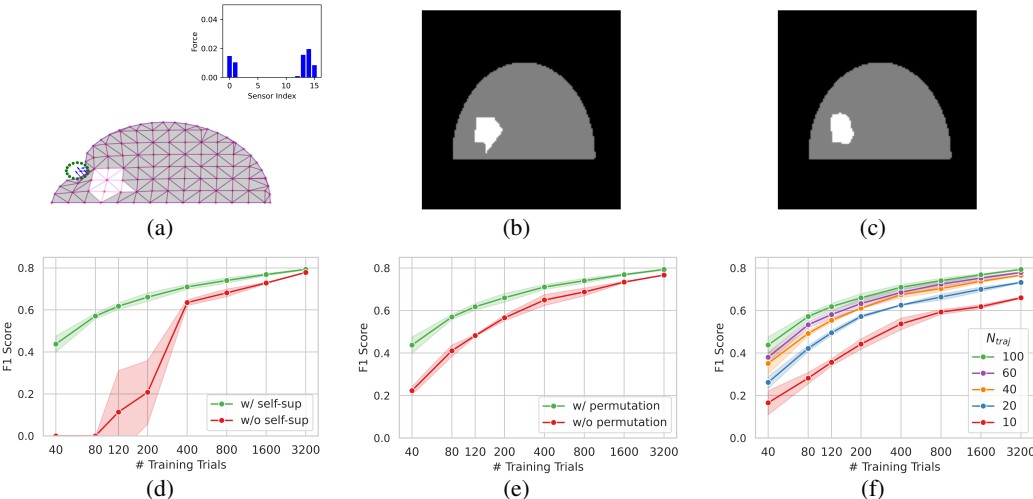

Figure 3: `PalpationSim` simulator and simulation results. (a) A 2-dimensional finite-element model of a round sensor pressing on a soft object with a harder lump inside; insert shows the forces on the sensor. (b+c) A ground truth image of the body (b), and a predicted image (c). (d-f) Image prediction results: (d) with and without self-supervised pretraining, (e) with and without permutation augmentation, (f) with different number of trajectories per trial. See text for details.

of the body and lump by different values of Poisson's ratio and Young's modulus [Lubliner and Papadopoulos, 2016]. The bottom vertices of the semi-circle are fixed in place, while the other vertices are free to move. A tactile sensor is modeled using 16 2-dimensional points arranged in a circle, with the center of the circle denoting its pose $x$ (here, $x$ is a 2-dimensional position instead of a 6-dimensional pose). Each point, when inside a triangular element, applies a spring-like repulsion force to the nearest edge of the triangle. Given $x$, we find the positions of all the vertices by minimizing the energy of the system using Adam [Kingma and Ba, 2014], and measure the forces on each sensor proportionally to their penetration into the body. Note that the simulation is *quasi-static* – per sensor position $x$, we measure the forces after the vertices have stabilized in a steady state. When we move the sensor, we measure the steady state of the system for each sensor position along the way (we warm-start the energy minimization with the previous solution for faster optimization). This design choice corresponds to a slow motion of the sensor in the real world.

The motions available to the probe $x_1, \ldots, x_T$ are linear trajectories that "press" on the soft body at various positions. The body observations $I$ are 2-dimensional $128 \times 128$ images where we discretize each pixel value into 3 classes: background, body, and lump, as depicted in Figures 3b,3c.

**Data Collection** We first randomly sample $N_p$ random bodies with different lump locations (if any), sizes, and variations in the Young modulus and Poisson ratio for each finite element. For each body, we collect $N_{trial} = 2$ "trials", where each trial contains a sequence of $N_{traj}$ trajectories pressing on the body from uniformly distributed angles. Between trials, we randomly modify the Poisson's ratio and Young's modulus for each finite element, and, in 10% of the models, we increase the size of the lump, for our change detection task. Full details for the data collection appear in Appendix B.2.

**Simulation Results** We focus on the following two questions: (1) how important is self-supervised pretraining, and (2) how do results scale with the number of models and the number of trajectories collected from each model. In addition, we report on a simple and useful augmentation.

Figures 3b and 3c demonstrate our image prediction results. In the following, we treat each pixel in the predicted image as a classification problem, and report the $F1$ score for the complete image [Manning et al., 2008], which we found to correlate well with visual image quality. We report results for different sizes of the data, per the total number of trials in the dataset (specifically, we train the models on a subsample of trials and corresponding images to report each data point in the figures). In Figure 3d we compare the image prediction using a pretrained representation (trained on the full data), as

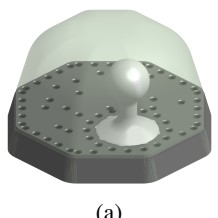 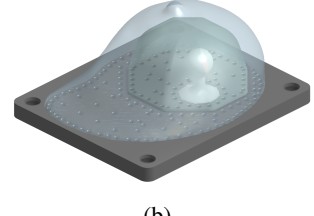 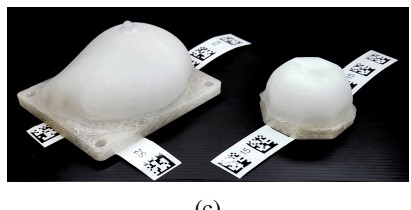

|(a)|(b)|(c)|

Figure 4: Modular Breast Phantom Design. (a) The *insert* has an octagonal 3D-printed base, a soft-silicone skin, and is filled with polyvinyl acetate slime. A soft silicone "lump" is embedded within and attached to the base. (b) The *shell* has two layers of soft silicone skin, with slime in between, where the bottom layer is attached to a 3D-printed base with an octagonal hole. The insert can be positioned in 8 possible orientations inside the shell. (c) An assembled shell+insert and a standalone insert. The bar-code labels allow to automatically record the component types and orientations using an overhead camera.

described in Section 3, with a supervised learning method that directly predicts the image from a sequence of measurements. Interestingly, pretraining improves results for all sizes of data, and is more dramatic when less data is available, showing the benefit of pretraining with large amounts of only tactile data. In Figure 3f, we show the scaling with respect to number of trajectories from each body used during pretraining; intuitively, performance plateaus when the trajectories sufficiently 'cover' the body. Finally, in Figure 3e we show that a simple augmentation of randomly changing the order of trajectories from the same trial during pretraining dramatically improves performance; we used this augmentation in all other reported results.

To conclude, the simulation study allowed us to dimension our real world study, and obtain a well-performing working point for our learning algorithms. We next report our real world results.

## 4.2 Real-World Results

Our goal is a proof-of-concept for artificial palpation. To this end, we designed an experimental procedure that, while artificial and simplistic in nature, addresses some of the realistic challenges in obtaining tactile measurements and ground truth imaging for breast palpation.

### 4.2.1 Benchmark Curation

**Modular Breast Phantoms:** At the core of our experiments is a novel breast phantom design (see Figure 4) that is composed of two modular parts, an outer *shell* and an *insert*, which can be assembled together in multiple orientations to collect data from a variety of bodies. Each part is composed of a soft silicone skin, and a polyvinyl acetate slime filling, mimicking the breast fat tissue. We add a lump of certain shape inside the insert made also of soft silicone. While our phantoms are clearly not anatomically accurate depictions of human breasts, we posit that they reflect a realistic range of tactile sensations present in breast palpation, as we verified with several physicians and breast oncologists. We provide a detailed description of the phantom and instructions for reproducing our design in Appendix A. We fabricated 6 shells with different slime characteristics and inserts with 8 different lumps positions and shapes, each in 3 different sizes, ranging from 8[mm] to 14[mm] diameter, which can be placed inside the shell in 8 different orientations, resulting in over 1150 possible different phantom combinations that we can collect data with.

**Automatic Palpation Data Collection:** We use a single XELA uSkin tactile sensor [Tomo et al., 2018], which has 30 3-dimensional force sensors, and records data at 85 Hz.[5]

To collect data in a consistent and repeatable manner, we use a Franka Emika Panda robotic manipulator with the tactile sensor attached to its end effector, as depicted in Figure 10. We control the arm using a hybrid force-motion controller [Lynch and Park, 2017] that applies a downward motion of the tip and a vertical force of 3.8N at a fixed orientation, implemented as a modification of the default force controller in panda-py [Elsner, 2023]. Such a downward motion at a particular

---

[5]We opted for the uSkin based on a preliminary comparison with a Digit sensor [Lambeta et al., 2020], reported in detail in Section A.1. We found the vision-based Digit to perform poorly inside the soft material, which we conjecture is due to the low spatial frequency of relevant forces, which the Digit is not optimized for.

$x - y$ position lasting for 5 seconds is termed a 'poke'. We execute pokes on a preset matrix of 110 positions, and for each poke we record both the sensor readings, the robot end effector poses, and their corresponding timestamps. In total, collecting data from a single phantom takes 20 minutes of robot time. Collecting data from $\sim 550$ phantom combinations resulted in a total of $\sim 60K$ pokes, and $\sim 30M$ instantaneous sensor readings. To mitigate time-varying bias in the Xela sensor, each measurement sequence was normalized relative its first sensor reading.

**Ground Truth Phantom MR Images:**   We generate ground truth labels for tactile imaging by scanning our phantoms in an MRI system. The motivation for this is threefold: (1) while we have 3D CAD designs for the phantom molds, our manufacturing process of the silicone skin has inaccuracies due to its manual nature; (2) the MRI scans, while not realistic due to the simple structure of our phantoms, are still susceptible to real-world challenges of noise and variations in positions and shape due to the softness of the phantom, making for a more challenging prediction problem; and (3) for a future human study, collecting ground truth data using MRI is a viable approach.

We scan our inserts[6] using a 3T MRI (Siemens Prisma) system with a 64-channel coil. The acquisition protocol involved a volumetric T2-weighted SPACE (fast spin echo [Bernstein et al., 2004]) acquisition. The parameters were a turbo factor of 270, resolution of $0.8 \times 0.8 \times 0.9$, field of view was $220 \times 200 \times 86.4mm$, TR of $3200ms$, and TE of $412ms$. The scans were accelerated by 4x using a standard undersampling technique. The images were reconstructed on the scanner using the GRAPPA algorithm [Griswold et al., 2002]. They were then imported to an external servers for training the networks. Since in our inserts lumps have similar height, for simplicity we predict a horizontal image slice instead of a complete 3D model (full details are in Section I).

### 4.2.2   Results

**Tactile Imaging:**   In Figure 5 we show several imaging results and their ground truth counterparts; a more extensive demonstration is provided in Section G. For comparison, we visualize a map of forces from the same data by taking the maximal recorded vertical force across sensors in each location. Evidently, our learning-based approach aggregates the scattered force measurements into a coherent object, as appears in the data, leading to a tactile image that is arguably more interpretable. For interpreting image quality, in addition to the F1 score, we consider two measures that evaluate the lump prediction accuracy: its area in the image (henceforth, size), and the position of its center of mass (CoM). In Table 1, we provide quantitative results, showing that our method (**Image Pred.**)

Table 1: Lump Size error, Center-of-Mass (CoM) error and F1 score. For prediction methods, we report standard deviation of the sample mean across 5 random seeds. For the average prediction, we report standard deviation across samples.

| Method | Size Error [%] ↓ | CoM Error [mm] ↓ | F1 Score [%] ↑ |
|---|---|---|---|
| Image Pred. | $23.0 \pm 2.1$ | $2.4 \pm 0.0$ | $74.4 \pm 0.1$ |
| Image Pred. ($0.5\times$ Data) | $20.9 \pm 0.9$ | $3.6 \pm 0.1$ | $65.3 \pm 0.1$ |
| Image Pred. ($0.25\times$ Data) | $41.6 \pm 3.2$ | $6.1 \pm 0.4$ | $43.9 \pm 2.6$ |
| MLP Pred. | $10.5 \pm 0.5$ | $2.9 \pm 0.1$ | - |
| Force Map + Image Pred. | $46.6 \pm 2.8$ | $5.3 \pm 0.1$ | $47.3 \pm 1.3$ |
| Average Pred. | $51.9 \pm 47.5$ | $12.1 \pm 4.1$ | - |

achieves 23% error in lump size and 2.4mm error in lump location. In comparison, the average errors for predicting the average lump area and location (**Average Pred.**) are 52% and 12.1mm, respectively. Therefore, our representation is clearly informative about the lump properties. We also trained a multilayer perceptron (MLP) predictor from our pretrained representation to $x - y$ CoM position and lump size (**MLP Pred.**). The lump size MLP was more accurate than the image predictor on size prediction, with 10.5% error. This shows that the representation contains even more information than is extracted by the image predictor, probably due to insufficient image data. We also evaluated a method that uses the (non-learnable) force map as a representation, with the same flow matching

---

[6]We scan inserts instead of the complete phantoms for time and cost reasons – we scan several inserts at once, and rotate the insert images digitally, obtaining scans for almost 200 different bodies. While removing the shell makes for a slightly easier prediction task, we found it challenging enough to obtain meaningful insights

image predictor as we used with our representation (**Force Map + Image Pred.**), see Section H for complete details. This method yielded significantly worse results, which strengthens the importance of our self-supervised learning approach.

As our approach is data-driven, we hypothesize that increasing the amount of data should improve the tactile imaging results. To support this, we repeat the self-supervised pre-training and MRI prediction with $0.5\times$ and $0.25\times$ less training data. As can be seen in Table 1, decreasing the amount of training data significantly worsens image prediction performance.

**Data Scaling:** Compared to non-learning approaches, our data-oriented paradigm benefits as we train on more data (as shown in Table 1). Still, collecting more labeled data (i.e. tactile measurements, coupled with an MRI scan) is costly. This problem will be even more significant for real human data. This issue raises a question - *does our approach scale with more unsupervised data only, without more labels?*

To test this we trained a model with 50% supervised data and 100% unsupervised data (and vice versa), the results are shown in Table 2. Adding unsupervised data alone resulted in over

|  | $0.5\times$ Sup. Data | $1\times$ Sup. Data |
|---|---|---|
| $0.5\times$ Rep. Data | $65.3 \pm 0.1$ | $74.5 \pm 1.7$ |
| $1\times$ Rep. Data | $72.7 \pm 0.3$ | $74.4 \pm 0.1$ |

Table 2: F1 score of our proposed tactile imaging approach with varying amount of unsupervised and supervise data. We report standard deviation of the sample mean across 3 random seeds.

$11\%$ increase of the F1 score, while adding the supervised data on top of it contributed only an additional $2\%$. In the context of a future study with human data, this result hints that costly labels can be replaced with easier to obtain self-supervised training.

**Change Detection:** To ground our results and highlight the task difficulty, we performed a small-scale human study for the change detection task. We asked $\sim 15$ participants to touch the phantom and detect whether or not we changed the insert to one with a larger lump at the same position (full technical details regarding the human study can be found in Section C). Each participant repeated the experiment $\sim 4$ times, resulting in a sample size of $\sim 60$. We compared with classification based on our lump size prediction for the same sample that was presented to the participants. We obtained a recall of $0.82$ and false-alarm-rate (FAR) of $0.19$, better than the human recall of $0.62$ and FAR of $0.32$. It is remarkable that these results were obtained with the Xela uSkin, for which the spatial density of sensors is almost two orders of magnitude smaller than the human finger [Corniani and Saal, 2020]. To further appreciate the task's difficulty, we provide a video in the supplementary material that visualizes how the lump moves inside the slime when touched.

**Shell Classification:** To further demonstrate the information content in our learned representation, we show in Section J of the supplementary material that it can be used to reliably predict the shell ID – information that is not available in the insert images. Thus, the representation is informative enough to capture small artifacts in the manufacturing process (all shells are manufactured in the same way), effectively distinguishing between the shells with an accuracy of $99.6 \pm 0.7$.

## 5  Discussion

Our results indicate that with enough data, a neural network trained to process tactile measurements yields a representation that is informative about the palpated body. Importantly, the performance of our method improves with additional data, without changing the hardware – unlike conventional tactile imaging methods that are not learning-based. How much, and what kind of, data is required to scale our approach to be clinically relevant?

Tactile measurements depend on the body being palpated, the sensor type, and the sensor movement. Relating to findings in other modalities such as vision and language, we predict that collecting massive tactile data from multiple sensors on general objects (not necessarily soft phantoms) may lead to foundation models for touch processing that can be fine-tuned to specific palpation tasks. The large dataset collected here can contribute to a collective data collection effort [O'Neill et al., 2024, Bell and Shimron, 2023]. One interesting question is whether language, which has played a key role

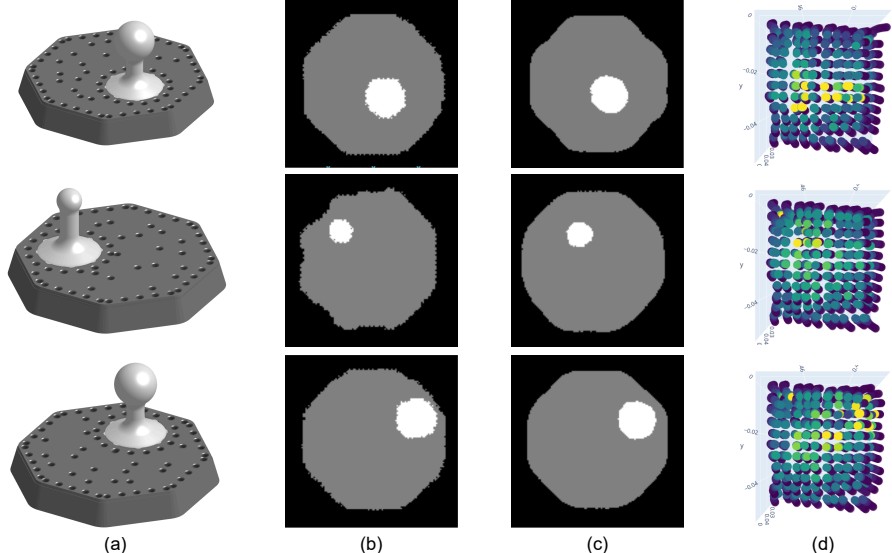

|  (a)  |  (b)  |  (c)  |  (d)  |

Figure 5: Tactile imaging with real data. Columns show: (a) 3D CAD design, (b) Ground-truth MRI image slice, (c) Predicted images using our method, (d) Force-map visualization, for 3 arbitrary bodies.

in foundation models for vision and robotics, is also important for touch (cf. the manual palpation instructions mentioned in our introduction).

Our data collection protocol requires a robot both for automation, but also for pose estimation. Extending our results to a human moving the sensor – a likely application, requires adjustments of the palpation trajectories we record to be more human-like or even apply active sensing [Scimeca et al., 2022], while pose estimation has standard solutions such as fiducial markers [Fiala, 2005].

Relating to our imaging results, MRI of human breast tissue reveals intricate details of small structures like ducts and blood vessels that do not exist in our fabricated models, making them significantly more complex to predict. In addition, the common use of contrast agent injections in breast scans to enhance the visibility of cancerous tumors is not accounted for in our current models. Thus, we cannot immediately deduce that our results will generalize to real human data. Nevertheless, as there is a correlation between abnormalities in breast tissue and their tactile sensing, we see promise in further investigating this direction. Moreover, although a vast body of work has explored deep learning for MRI [Heckel et al., 2024], to the best of our knowledge, learning the mapping from sensory data to MR images has not been explored yet, and our work initiates this new line of research.

Finally, in addition to the technological challenges in artificial palpation, there are clinical and sociological challenges. While several clinical trials have been performed with conventional tactile imaging [Bexa Inc., 2024], we do not yet have information about the sensitivity and specificity of learning based approaches, or palpation using low-cost sensors. On a positive note, the survey by Jenkinson et al. [2023a] showed a generally positive reaction to automated breast cancer screening.

# 6   Acknowledgements

This work received funding from the European Union (ERC, Bayes-RL, Project Number 101041250). Views and opinions expressed are however those of the authors only and do not necessarily reflect those of the European Union or the European Research Council Executive Agency. Neither the European Union nor the granting authority can be held responsible for them. E.S. is a Horev Fellow and acknowledges funding support from the Technion's Leaders in Science and Technology program and Alon Fellowship. We are grateful for the support of the May-Dahl-Blum Technion Human MRI research center staff and services.

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

# A    Robotic Data Collection Setup

While the simulation setup is useful for quick development iterations, it does not mimic the real-world, and the learned representations will fail to generalize outside the simulation. To test our approach on a real tactile sensor we designed a setup to mimic a simplistic palpation examination.

## A.1    Xela uSkin and Digit Comparison

We use a single Xela uSkin [Tomo et al., 2018] which has 30 force sensors, each measuring forces in $x$, $y$, and $z$ directions.

In our work, we have not made strong assumptions on the tactile sensor, and in theory, our learning approach can be used with other sensors. We have tested the behavior of a Digit sensor [Lambeta et al., 2020]. To test the behavior of Digit compared to the uSkin, we performed a simple experiment. We first touched a phantom containing an insert without a lump, and then touched an insert with a lump exactly on top of the lump. We repeated this test with the Digit and uSkin, and the results are reported in Figure 6. As can be seen in the Figure, when touching the samples with Xela sensor, we get significant measurements when touching the shell, while the digit doesn't seem to be affected by it. We tried to measure various objects with the digit sensor and saw that it is very good at detecting surface patterns and is responsive to hard surfaces, but is very lacking in sensitivity to soft objects and depth changes. In an experiment we conducted earlier, we tried the Xela vs. the digit in a previous generation of inserts. We easily classified the touching lump / no lump with the Xela and completely failed with the digit.

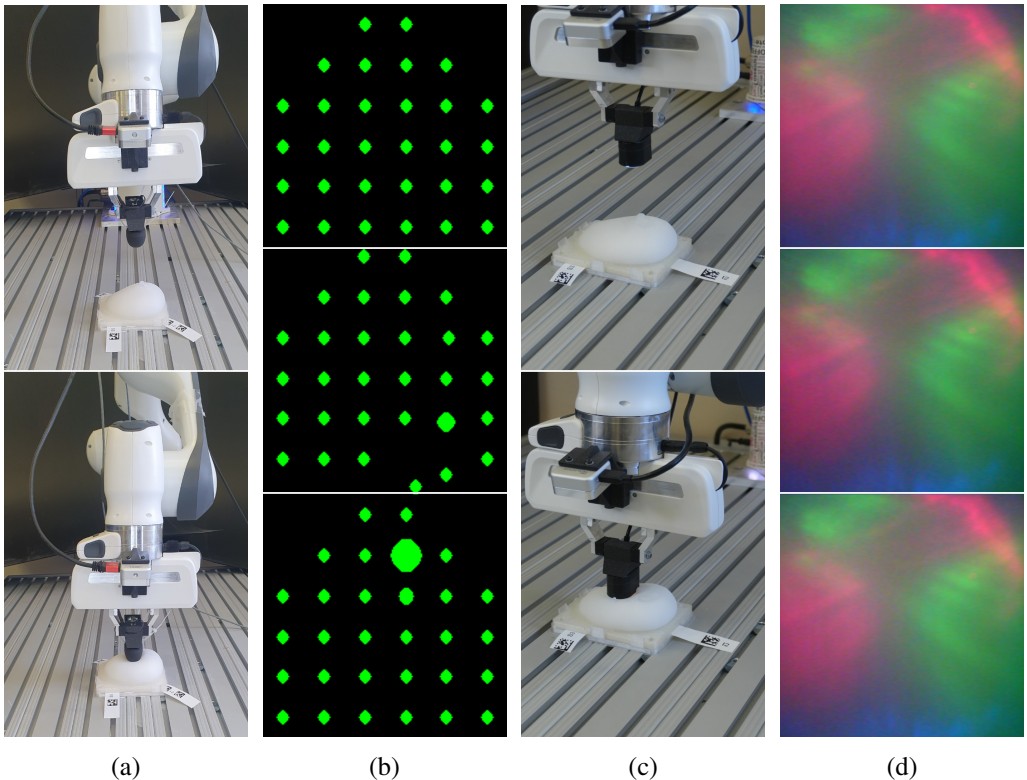

(a)        (b)        (c)        (d)

Figure 6: Comparison of tactile perception pipelines. (a), (b): Xela system — robot and tactile sensor views. (b) shows captures from the Xela visualizer. The top image depicts a state of no touching, the middle one touching a phantom without a lump, and the bottom one touching a lump. (c), (d): Digit system — robot and tactile sensor views. (d) Images obtained by the Digit sensor. The sensor positions are the same as in (b).

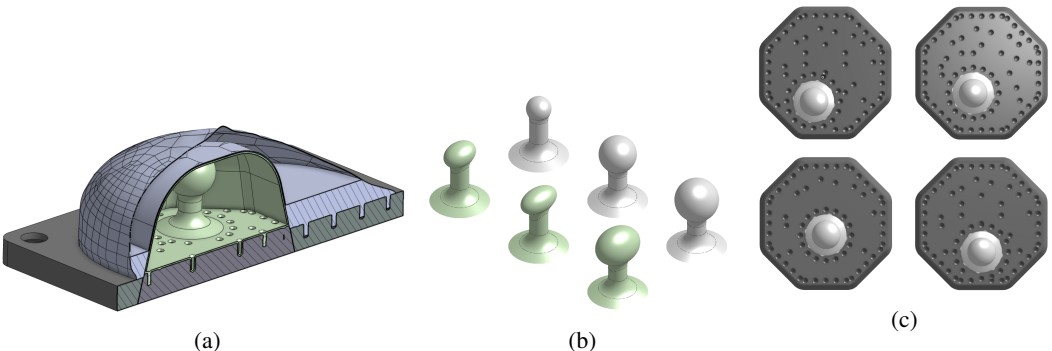

(a)                                (b)                                (c)

Figure 7: In (a): Section view of the shell and insert assembly. In green, insert silicone skin. In blue, shell silicone skin. In dark gray, PLA bases for the insert and shell. Voids are filled with slime. Note that holes in the bases are filled with silicone for anchoring. (b) Lump sizes: In white from furthest to closest, 8, 12, and 14mm diameter spheres. In Green ellipsoids with long/short axes of 12/8, 14/8, and 16/12mm. In (c), four insert bases, each with a 14mm lump at a different location. Other lumps (not shown) can be inserted at the same locations.

## A.2 Breast Phantoms

For the breast phantoms, we aimed for a modular design, which will allow for effective data collection. Our phantoms are built from two parts - a shell and an insert. Both shell and insert are made using a thin outer layer of silicone (Smooth On Dragonskin FX) filled with "slime" - a gel matrix made of polyvinyl acetate (PVAc), borax, and water. We shall refer to the outer layer as skin. The skin is attached to a base plate made of polylactic acid (PLA) plastic using pre-made holes in the base plate. Inside the insert, we have the lump, which is made of the same Dragonskin FX silicone as the outer layer. See Figure 7a for a section view of the assembly. The insert has an octagon-shaped base plate that fits into the shell's base plate, this is so we can have eight configurations for each insert, lessening the amount of inserts needed. We created 24 inserts in the following configuration: 4 lump locations as seen in Figure 7c, with 3 spherical and 3 ellipsoidal lump sizes as seen in Figure 7b.

## A.3 Phantoms Fabrication

The skin for both the insert, and shell is constructed much like hollow chocolate bunnies or Easter eggs: we pour a small amount of silicone into a multipart mold and brush all the surfaces (Figure 8a), next we close the mold and rotate it until it sets. When we open the mold, we have a hollow shell or insert. During the molding process the silicone enters holes in the base, which serve as anchor points for the silicone as seen in Figure 7a. For the shell, the process with silicone ends at this stage. In the case of the insert, we have to remove the lump mold (Figures 8b and 8c), pour silicone into the lump skin, and close the hole in the base with a plug (Figure 8d).

Following the silicone stage, we then inject the slime using a needle through a thick part of the phantom (Figure 8e); this works for both shell and insert. This concludes fabrication of the phantom.

## A.4 Transparent Insert

To showcase the task's difficulty, we manufactured a single transparent insert so that the lump will be visible during handling. As can be seen in Figure 9, the lump moves when touched. This is a realistic phenomenon of the 3D structure that would appear in real tissue.

## A.5 Data Collection Setup Design

To collect data in a consistent and repeatable manner, we utilize a Franka Emika Panda robotic arm with 7-DoF. The complete setup, which can be seen in Figure 10, consists of the robotic manipulator, the tactile sensor held as the end effector, a camera attached to the manipulator, and the phantom. Both are attached/held using 3d printed adapters. We controlled the panda using panda-py [Elsner, 2023]. For the actual data sampling, we edited the force controller to be a force-position controller

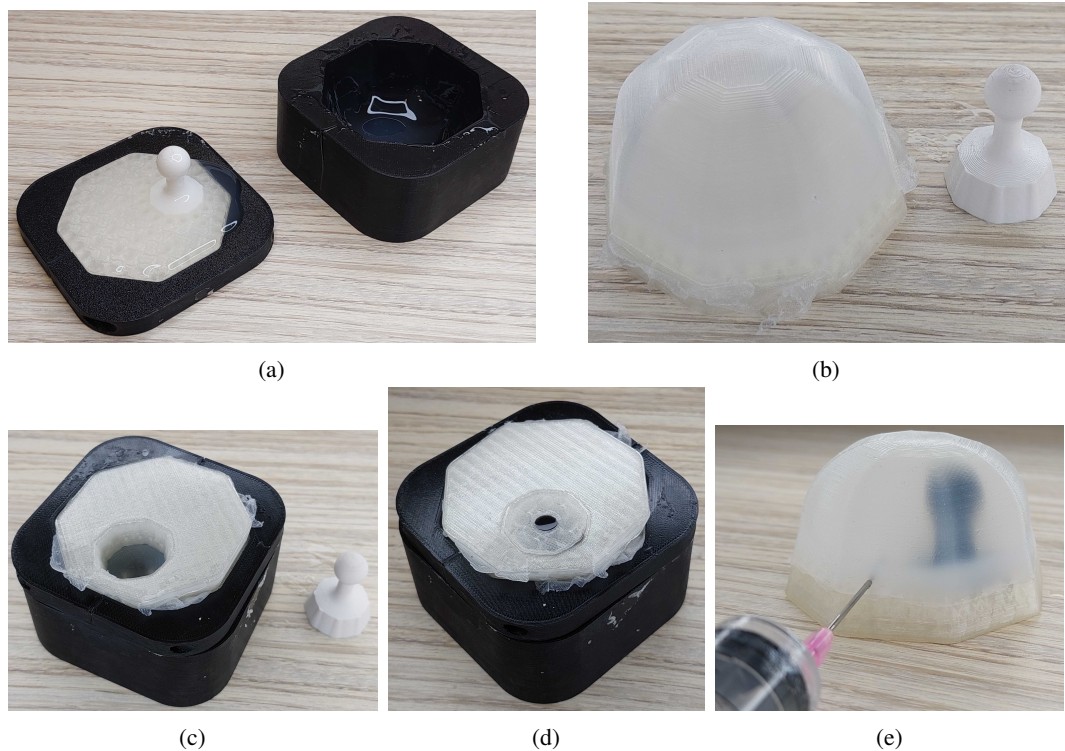

Figure 8: Fabrication process for insert. (a) Silicone on the mold parts is fully covered. (b) Parts after removal from the mold. (c) using mold as support. (d) Hole filled with silicone. The silicone is colored black for illustration purposes. (e) Injection of slime.

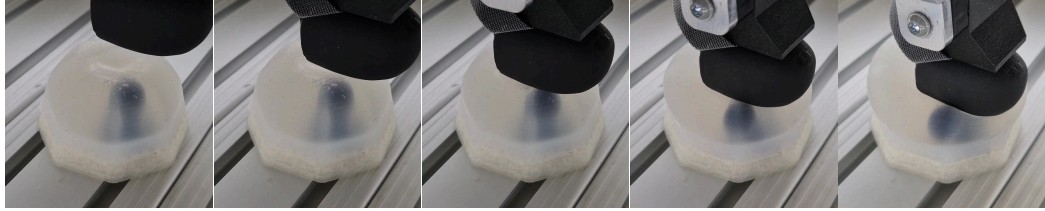

Figure 9: Poke trajectory on a special transparent insert.

in order to press the phantom with a specific force at a specific point. In order to be organized and automatic as possible, we used the camera to scan data matrices (similar to QR codes) that are attached to the shells and inserts to automatically record the used items and the insert orientation. Data was recorded for trajectories at 110 points on the phantom. We recorded Xela data at $85[Hz]$, robot data at $100[Hz]$ and RGB images from the camera at $10[Hz]$. We collected the data with a fixed force and fixed sensor orientation.

### A.6 Data Collection Technical Details

The robotic manipulator first scans the attached data matrices (similar to QR codes) to determine the labels of the phantom and the insert, and also to obtain the orientation of the insert inside the phantom. The robot goes to each $x, y$ position (110 trajectories) and starts to descend in a straight line in $z$ by 2[mm] steps until we sense a significant force with the Xela sensor (the sensor output is noisy), and then backtracks 2[mm]. after that we use a force/position controller that controls the force in z and the position in $x, y$ of the end effector for 5 seconds, during that time we record all the Xela measurements ( 85[Hz]), data from the robot (100[Hz] $q$, $dq$, calculated force, $tau\_J$,$O\_T\_EE$ ) and RGB images from the realsense camera (10[Hz] currently not in use). We considered using the depth capability of the RealSense camera, but it had too much noise in these settings. We add

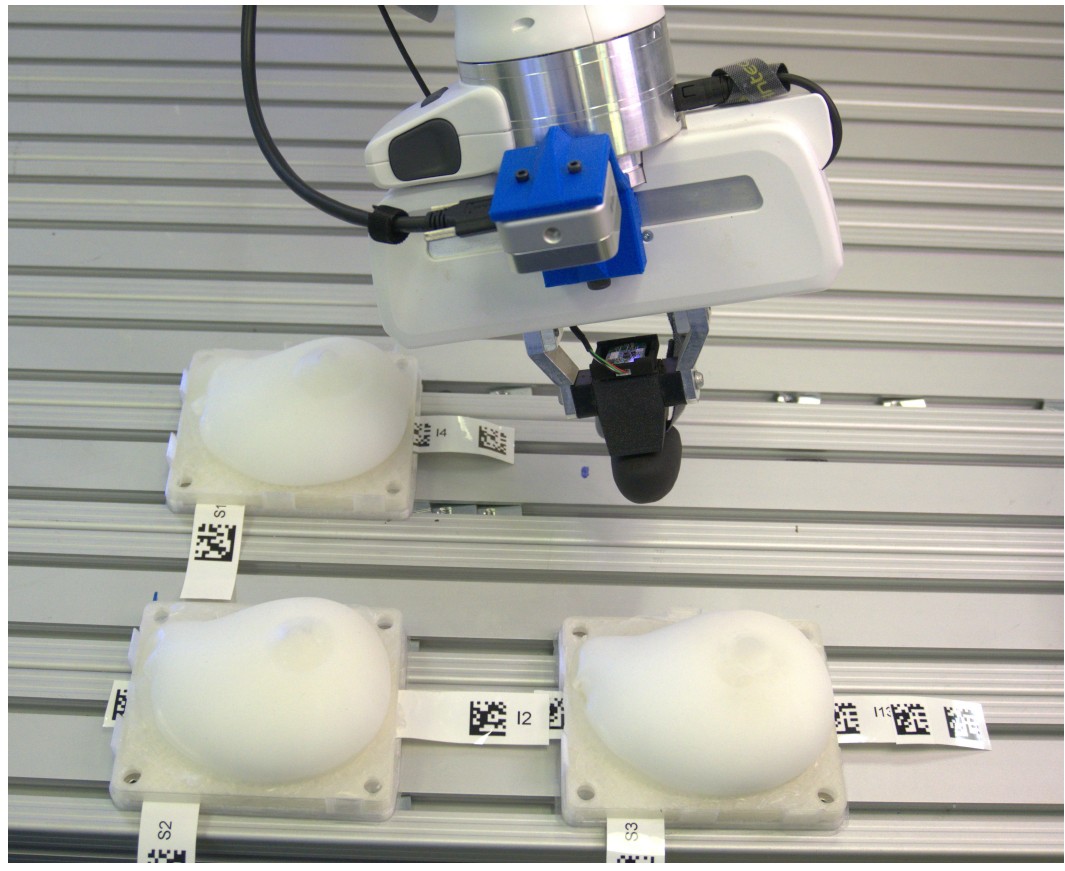

Figure 10: Our automatic robotic data collection setup consists of our breast phantom (including visible QR codes), a Franka Emika Panda robotic arm with a mounted Xela uSkin tactile sensor, and an RGB camera.

the relevant timestamps to all the data. Robot data consists of the libfranka variables q, dq - joint angles and angular velocity, calculated force - force and torque at the end effector calculated using the Jacobian, $\tau_J$ - Measured link-side joint torque sensor signals, and $O\_T\_EE$ - Measured end effector pose in base frame. After testing a number of positions/sensor angles/forces, we decided that the best scan for our needs, which takes 20 minutes, will consist of 110 positions on the phantom with a single fixed force 3.8[N] and a single sensor orientation.

## B    PalpationSim

### B.1    Simulation Visualization

In Figure 11 we visualize trajectories from two models in the simulation. As can be seen, the lump inside the breast model affects the sensed force, as expected.

### B.2    Simulation Data Collection Hyperparameters

The hyperparameters used for the simulation data collection are shown in Table 3.

### B.3    Simulation Imaging Results

To visualize the results shown in Section 4, we present in Figure 12 20 randomly sampled model-image reconstruction results in the simulation. As we can see our image predictor can reconstruct the model image fairly well from our tactile sequence representation.

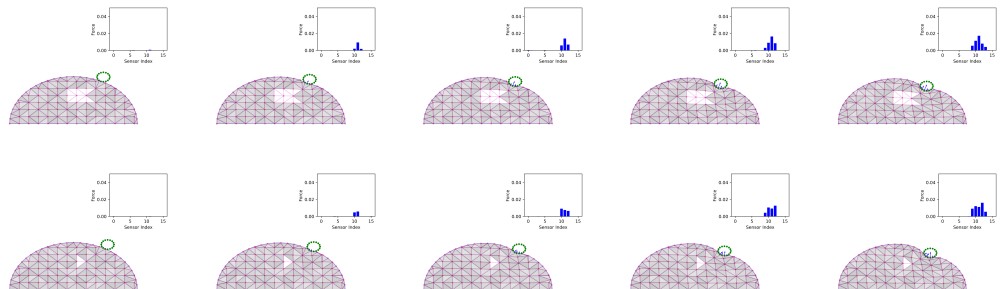

Figure 11: Comparison of selected simulation trajectories for big and small lumps. The simulation interface includes a force graph measured by the probe (including force vectors for easy visualization), the probe in green, and the breast model.

| Name | Value |
|---|---|
| $\sigma_{noise}$ | 0.0001 |
| $N_{points}$ | 16 |
| $R_{probe}$ | 0.1 |
| $\kappa_{collision}$ | 0.01 |

(a) Probe hyperparameters

| Name | Value |
|---|---|
| $\beta_1$ | 0.2 |
| $\beta_2$ | 0.999 |
| $lr$ | 0.001 |

(b) Adam optimizer hyperparameters

| Name | Value |
|---|---|
| $R_{Model}$ | $\mathcal{N}(1, 0.01)$ |
| $L_{grid}$ | 0.15 |
| $L_{perimeter}$ | 0.1 |
| $N_{points}$ | 0.001 |
| $\mu_{ym}$ | $U[0.0027, 0.0033]$ |
| $\sigma_{ym}$ | 0.0002 |
| $\mu_{pr}$ | $U[0.09, 0.11]$ |
| $\sigma_{pr}$ | 0.01 |

(c) Breast-model hyperparameters

| Name | Value |
|---|---|
| $ym_{lump}$ | 0.01 |
| $pr_{lump}$ | 0.1 |
| $p_{change}$ | 0.1 |
| $center_{lump}$ | $Ring(0.44, 0.55)$ |
| $R_{lump-change}$ | $U[0.07, 0.15]$ |
| $R_{lump-no-change}$ | $U[0.11, 0.21]$ |
| $\Delta_{R_{lump}}$ | $\mathcal{N}(0.03, 0.01)$ |

(d) Lump hyperparameters

Table 3: Hyperparameters used for the simulation data collection

## B.4 Simulation Change Detection

Since our image reconstruction in simulation accurately predicts the lump shape and location, we can use the image prediction for change detection and localization. As can be seen in Figure 13, given two trials from the same model, we first generate multiple image predictions by taking a different permutation of the trial trajectories (we used 10 permutations in practice). Using the multiple prediction, we can generate a joint confidence map:

$$c[i] = \frac{C}{C + \frac{1}{2}\left(\sigma_1[i] + \sigma_2[i]\right)}$$

Where $c[i]$, $\sigma_1[i]$ and $\sigma_2[i]$ are the confidence map, first trial standard deviation and second trial standard deviation at pixel $i, \forall i \in [1, 128 \cdot 128]$, respectively (the standard deviations are calculated across the different predicted image of each trial). The confidence calculation behaves such that very high and low standard deviations correspond to a confidence of 0 and 1, respectively, and when $\sigma_1[i] = \sigma_2[i] = C$ we get $c[i] = \frac{1}{2}$. Finally, each pixel gets a score:

$$s[i] = |\mu_1[i] - \mu_2[i]| \cdot c[i]$$

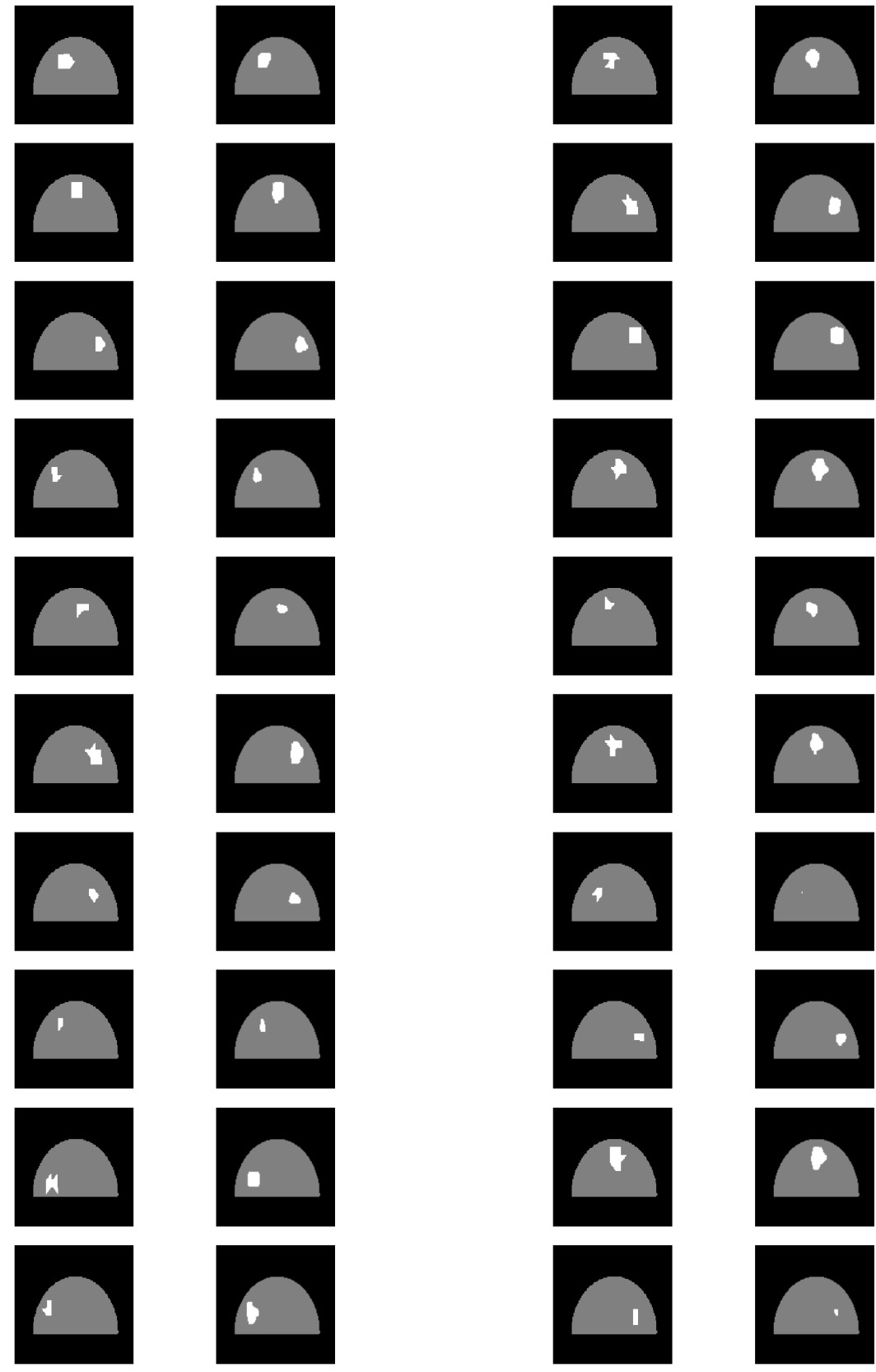

Figure 12: 20 random test results of our simulation tactile imaging predictions. The ground-truth model image and our prediction are on the left and right of each column, respectively.

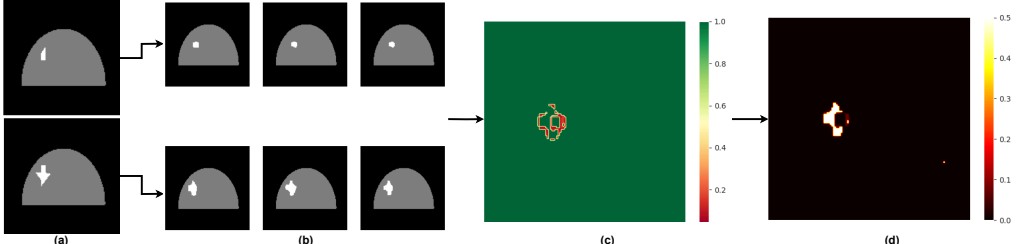

Figure 13: Our simulation change detection algorithm. (a) Two ground truth model images (b) Prediction of multiple images using the perturbations augmentation from each of the trials (c) Generation of a confidence map (d) Change map generation

Where $\mu_1$ and $\mu_2$ are the average pixel value at pixel $i, \forall i \in [1, 128 \cdot 128]$. Using the scores we can plot a change score map and localize changes.

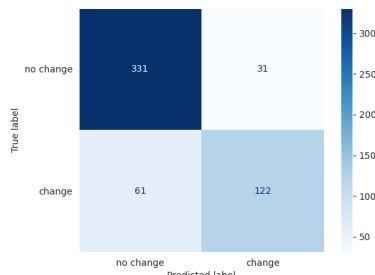

Figure 14: Our change prediction confusion matrix, when setting the threshold on the change score to 0.1

To aggregate the change map to a single score for the two trials, we simply take the mean over all pixels. We report the confusion matrix when setting the threshold to 0.1 in Figure 14.

## C Human Study Procedure and Results

**Procedure** The full procedure and given instructions for each participant are detailed below:

For explanation purposes, a single insert is put into a single shell.

> Our research is about detecting breast cancer using artificial palpation. We have breast phantoms, consisting of shells with inserts inside them, where each insert has a lump inside it. All lumps are spherical, and change in diameter and location. There is always a single lump inside of the insert. Your task will be to try and detect changes in the lump after a given amount certain amount of time. The lump can only grow larger or stay the same; it cannot change location, nor shrink. The lump will change with a 0.5 probablity, otherwise it will stay the same. Please follow these instructions while palpating:
>
> - Use only one finger when palpating
> - You can press however you like and for as long as you want to, but use reasonable force not to tear the phantom
> - In your answer, state "change" or "no change"
>
> Here, there is a phantom with a lump inside, just as an example, it is not part of the trial, you can feel it to get a sense of the task.

Next, the insert is changed, with the smallest possible change in the lump size (to calibrate the participants answers).

> Now we have replaced the insert to have a larger lump size, such that the change in size is the lowest possible change. You can now palpate to understand what a change might feel like.

Finally, the following steps are repeated $\sim 4$ times:

1. A completely new insert is put into a phantom (with a random lump location, orientation, and size)

2. The participant palpates the phantom

3. After the time interval, the lump, with 0.5 probability the insert is replaced with a different insert with the same location and orientation, but a larger size, and with 0.5 probability, the insert is taken out and put back in.

4. The participant palpates the phantom again, and their answer is logged together with the experiment metadata.

**Results** We had $\sim 15$ participants, and as explained above, each participant repeated the experiment $\sim 4$ times (with different time intervals). The results were 23/34 ($\sim 68\%$) correct non-changing classifications and 18/29 ($\sim 62\%$)correct changing classifications.

|  | Humans | Ours |
|---|---|---|
| **FAR** $\downarrow$ | 0.32 | 0.19 |
| **Recall** $\uparrow$ | 0.62 | 0.82 |

Table 4: Change detection recall and FAR for human participants and our approach.

**Ethical Concerns** The human study was approved by the Technion's Institutional Review Board (IRB), approval number 2025-068. There are no dangers to participants. The experiment is equivalent to playing with a squishy toy for several minutes. The materials are safe – silicone, and the inner filling (which is not touched by participants) made out of gel that is commonly sold as a children's toy (Borax, water, and glue). There is absolutely no inconvenience of any kind to the participants during the experiment. This experiment allows us to calibrate the performance of our system in comparison to human skills, which is important for understanding the potential of our research. Each participant received financial compensation above the minimum wage. The data recorded is kept private on a secure server, which only contains the names and emails of the participants and answers to questions. All files are password-protected, with passwords only known to the relevant researchers. The results we publish are summarized statistics, and do not contain any personal information about participants. Each participant interacted directly with one of the researchers in the project. We explained verbally and in written instructions to participants that their participation is voluntary and that they can withdraw at any time with no negative consequences for them.

## D  Self-supervised Training Technical Details

Our self-supervised architecture, as shown in Figure 2, is composed of the FLE (Force Location Encoder), a sequence encoder, and a force decoder. The FLE is simply adding a linear projection for the forces and a basic sinusoidal Positional Encoding (PE) for the 6-dim locations (both the projection and the PE are of size 256). The sequence encoder is a one layer RNN with a GRU and with a 1024 hidden size. The force decoder first linearly projects the input representation and uses PE on the desired reconstruction to size 1020, and adds both together. Next, a three-layer MLP is used (with a 2048, 1024 hidden sizes) to predict the forces at the desired location. An MSE loss (with random indices sampling) is used as shown in Section 3.2.

A visualization of the results of the reconstruction from the last representation is shown in Figure 15.

## E  Downstream Tasks Technical Details

To reconstruct the MRI images, we use the pre-trained (and frozen) encoder from the self-supervised step. We aim to learn a mapping from the last-step representation $z_T$ to the model MRI image. To do so, we use a vanilla conditional flow matching to reconstruct the $128 \times 128$ image from the vector representation. After training, in order to sample an image, we simply draw a single noise sample

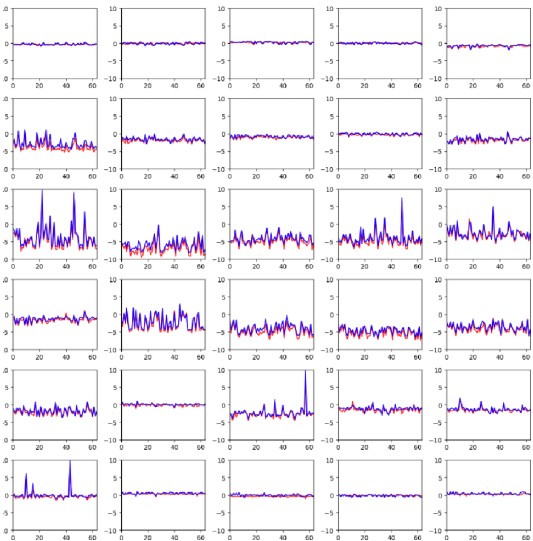

Figure 15: Force reconstruction on real data of the $z$-axis component of each of the 30 sensors from the representation in the last time step ($z_N$). In blue and red are the true and reconstructed forces, respectively.

and map it using the learned conditional mapping to an image. For the tactile imaging problem, we randomly split the trials to train and test (with the same split for the representation learning and the imaging). As for the change detection, we split per insert configuration, where all trials from the same configuration are either in train or test (this is done to have negative examples in the change detection task).

## F   Architecture Ablation

To motivate the chosen architectures, we perform an ablation study, both for the representation learning and for the tactile imaging. For the representation learning architecture we tested both an RNN with GRU (**GRU**) and a two-layer vanilla transformer (**TR**). For both architectures, we also considered a version where some of the input poke trajectories (e.g 20%) are masked, and we reconstruct only these. For the tactile imaging architecture, we considered the flow matching (**FM**) and a series of transposed convolutions (**TC**). We perform the training for the ablation study using $\times 0.5$ of our full training data due to cost and time concerns.

The results of the ablation study can be seen in Table 5. While this ablation study is preliminary, we chose the GRU and flow matching models for the representation learning and tactile imaging, respectively. We leave further investigation into more advanced architecture to future work.

Table 5: Lump Size and Center-of-Mass (CoM) errors. We report standard deviation of the sample mean across 3 random seeds.

| Method | Size Error [%] ↓ | CoM Error [mm] ↓ |
| --- | --- | --- |
| GRU + FM (Ours) | $20.9 \pm 0.9$ | $3.6 \pm 0.1$ |
| GRU + TC | $27.9 \pm 1.8$ | $3.5 \pm 0.1$ |
| TR + FM | $35.4 \pm 0.7$ | $6.7 \pm 0.3$ |
| Masked TR + FM | $57.9 \pm 2.0$ | $9.9 \pm 0.1$ |
| Masked GRU + FM | $23.9 \pm 1.0$ | $4.0 \pm 0.1$ |

# G   Tactile Imaging Results

To visualize the results shown in Table 1, we present in Figure 16 40 randomly sampled MRI prediction results. The predictions are accurate, in terms of lump location and size, as was quantitatively shown in Table 1.

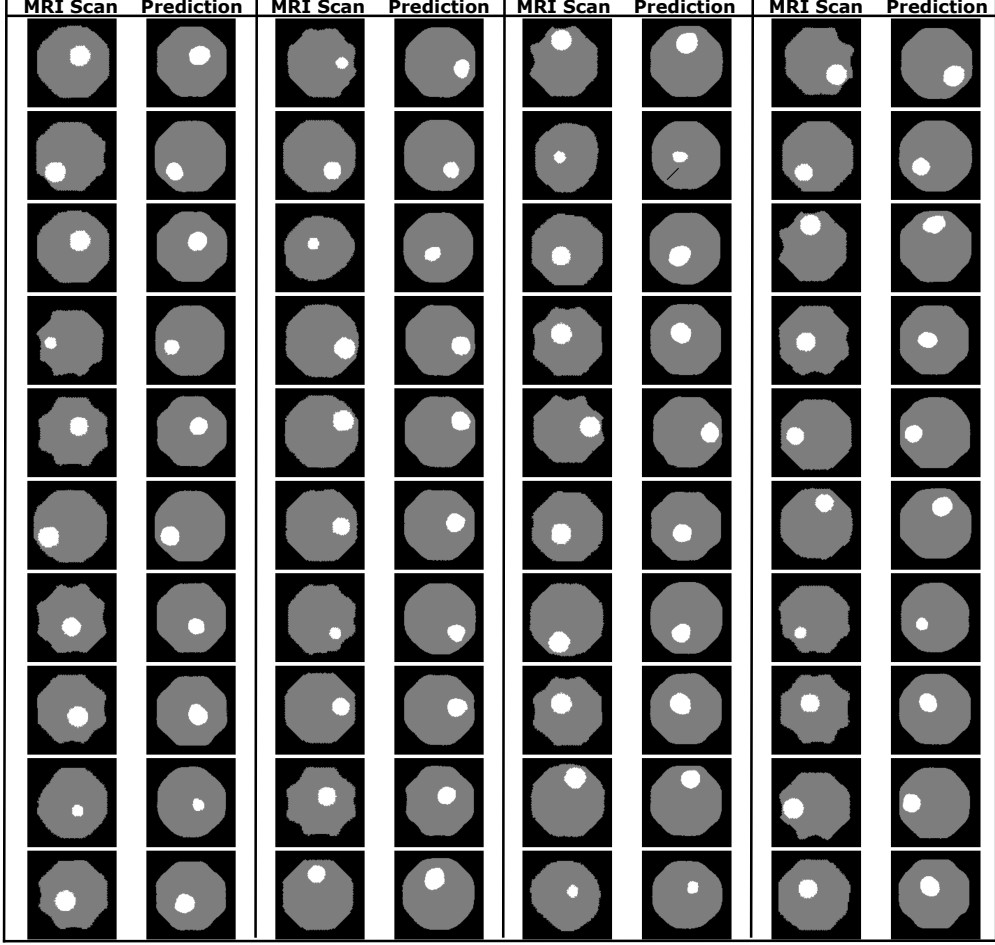

Figure 16: 40 random test results of our MRI predictions. The ground-truth MRI image and our prediction are on the left and right of each column, respectively.

## H  Force Map Generation Procedure

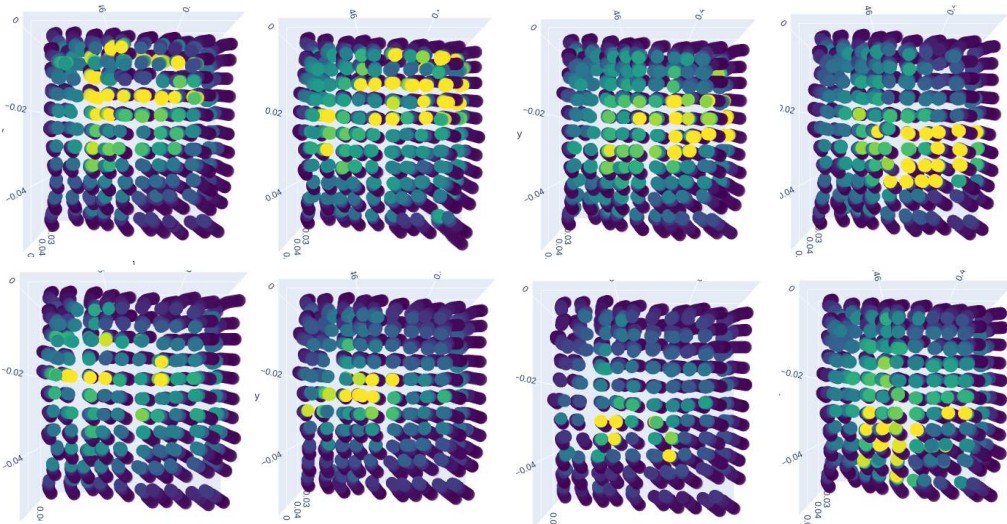

Figure 17: Top view of our 3D tactile visualization for a constant insert. Each visualization corresponds to a trial taken from the insert at a different orientation (from rotation 1 to 8, clockwise, starting from 1 in the top left corner).

The fact that tactile measurements are not interpretable in their raw form, makes it difficult to debug the data collection process. To this end we created a visualization tool for tactile data, collected from a single trial (with multiple trajectories). We aggregate the tactile measurements at each location by taking the maximal $z$ force measured by all 30 sensors. The result can be seen in Figure 17, and an interactive 3D version can be found in the supplementary materials.

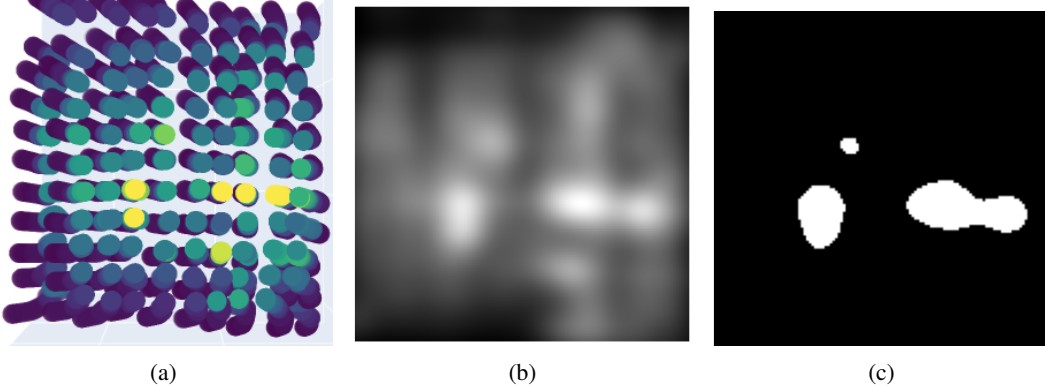

|       (a)       |       (b)       |       (c)       |

Figure 18: Force map generation process. (a) The 3D forces visualization. (b) The KDE image. (c) The resulting image after taking a threshold over the KDE image.

As can be seen in Figure 17, the generated force map is correlated to the lump location. Hence, we hypothesized that it can be used as a baseline for the tactile-imaging problem. As a first simple baseline, we take the mean over the forces and locations of the last $10\%$ data points of each trajectory in the 3D map. Next, we use Kernel Density Estimation (KDE) to produce an image from individual measurements. Finally, we take a threshold to binarize the image (which we set to maximize test accuracy). The result can be seen in Figure 18. As can be seen from the image, although the force map is very informative, it is not enough for accurate lump size and CoM prediction.

The result above illustrates that we cannot *directly* use the force map to predict the lump properties. To further motivate our approach, we also show that the force map is an *insufficient representation* for

tactile imaging. To do so, we trained an identical flow matching predictor to the one we used on top of our learned representation, but with a flattened force map. The results can be seen in 1. Clearly, the (non-learnable) force-map representation underperforms compared to our approach. We have also tried alternative architectures for the prediction, including a convolutional architecture on top of a non-flatten force-map representation.

## I  MRI Image Preprocessing

As explained in the main text, our imaging reconstruction is a classification problem, aiming to reconstruct one of the classes - background/body/lump for each pixel. So, in order to use the MRI images as ground-truth, we first had to build a pre-processing tool.

Although MRI data is 3D in nature, since our lumps are manufactured with similar heights, we chose to work with a single horizontal slice for simplicity. The procedure for our MRI pre-processing is shown in Figure 19, and examples of the final result are shown in Figure 20

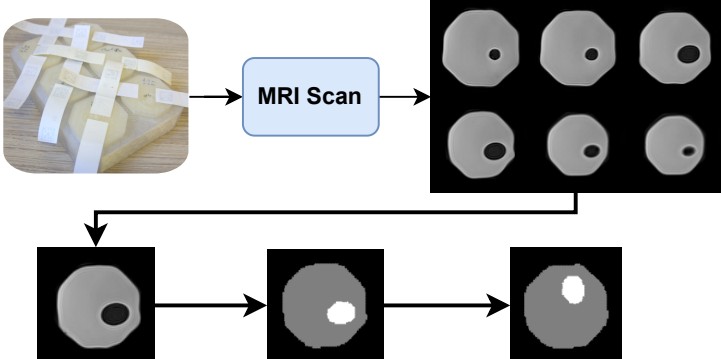

Figure 19: MRI pre-processing procedure. We 3D printed a hub, which can hold up to 6 inserts at a time for the MRI scan in order to save time and resources. After the MRI scan, we take a slice at a constant height and use Otsu's threshold Otsu et al. [1975] to binarize the image. Finally, we can rotate the scan to achieve any desired insert orientation.

## J  Shell Classification

To further show the expressiveness of our extracted representation, we chose another downstream task that cannot be inferred from the tactile imaging output. We aimed to classify the shell based on the tactile sequence. We have trained a small MLP on top of the frozen representation with a cross-entropy loss, aiming to classify between the 4 possible shells. Although the shells were manufactured in the exact same way, the representation is expressive enough to capture small manufacturing artifacts, effectively distinguishing between the shells with an accuracy of $99.6 \pm 0.7$ on test sequences. The confusion matrix can be seen in Figure 21 .

## K  Compute

The PalpationSim simulator runs on CPU. We collected data from it by running the simulation on $\sim 500$ CPUs to parallelize the process, but a single instance is very light-weight. All of the training procedures, including the self-supervised phase and the image reconstruction training ran on a cluster of 12 A4000 GPUs, although each single run needed only a single GPU to run.

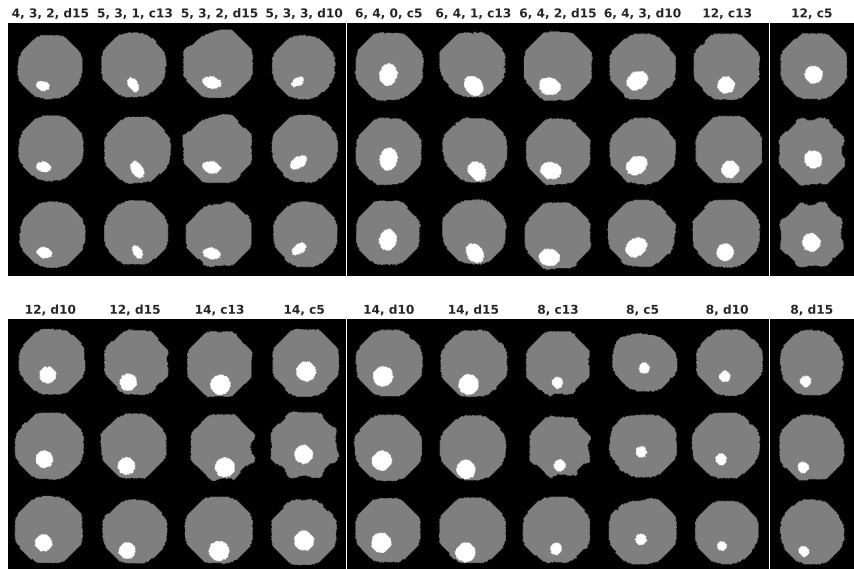

Figure 20: MRI images used as ground-truth, after pre-processing. Each insert was scanned 3 times. Above each trio, we show the first and second radii, the orientation, and location (either diagonal of center and radius from center) of the ellipsoid lump, where, in the case of a spherical lump, only a signed radius and orientations are shown.

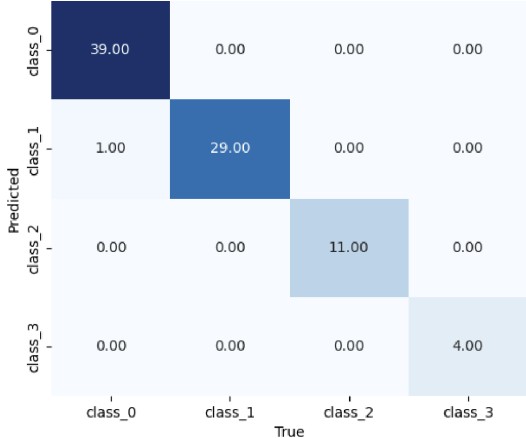

Figure 21: Confusion matrix for the shell classification downstream task on test sequences. Other than one mistake, the model has successfully classified all sequences.

