# OpenReview forum: "Toward Artificial Palpation: Representation Learning of Touch on Soft Bodies"
_NeurIPS.cc/2025/Conference — NeurIPS 2025 poster_

### Official Review · Reviewer_SEnm · 2025-06-27

**Clarity:** 3
**Significance:** 3
**Originality:** 2
**Rating:** 5
**Confidence:** 4

**Summary:**

The paper presents a pipeline to train tactile representations from trajectories of poking either a simulated or real breast phantom with embedded lumps of different sizes and locations. For this, the authors built a custom FEM-based simulator, PalpationSim as well as an automated data collection pipeline in the real world based on a Franka Panda arm and a Xela uSkin sensor. They furthermore introduce the manufacturing process of artificial breast phantoms for data collection.

The tactile representations are learned in a self-supervised way by predicting the next or previous tactile measurement of a poking trajectory. The paper shows that in the FEM-based simulated environment, the learned tactile representations are very informative for downstream tasks such as predicting a map of the breast phantom (based on MRI images). The authors also present results on data collected with real breast phantoms and show a comparison to human performance with a human subject study.

**Questions:**

How can the simulation results and the real world results be related? Which conclusions from the simulation can be used for the real world?
The authors state that the simulation was developed among other things to evaluate the amount of data necessary to obtain meaningful results. However, this objective does not seem to be met, as the authors describe severe overfitting of the image prediction network on real data, however, with the simulated data this is not the case. It would make the paper more impactful if it was more clear, which findings of the simulation can be transferred to the real world, as the simulated experimental results are significantly better than the one on the real phantom.

In the real-world experiments, the authors conduct a baseline experiment on the raw force maps that does not involve any learning, but instead a thresholding operation and report this to be much worse. A better baseline in my opinion would be to compare the learning-based method on the force maps as a representation, instead of the self-supervised learned one. This would also give conclusions about whether pretraining helps for the real-world experiments and allow a real assessment of the main claim of the paper, namely that the new representations are more informative than force maps
Other design choices, such as indenter shape or number of force sensors in the simulation, should be justified and investigated.

The data collection procedure in simulation and the real world seems to differ significantly. In simulation, the indenter is approaching the soft body from a normal direction, while in the real world, the sensor is pressed down from the z direction, leading the different approaching angles relative to the surface at every location. How does this major difference impact the comparability of the results?

In the method formulation, a tactile sensor is described as a rigid body. However, many of the cited vision-based tactile sensors are soft. How high does the difference in material properties have to be, to assume that the tactile sensor is a rigid body? Would it change the theoretical background if the sensor were a soft body? The paper is presenting a more general theoretical framework in the beginning, therefore it would be interesting under which constraints this holds.

The architecture choices of using a GRU over a transformer-architecture are not transparent to me. Earlier results are mentioned but not reported. According to the manuscript, the GRU was chosen as it scales more easily to long sequences, while transformers are known to be better at processing longer sequences without the problem of vanishing gradients. Reporting the initial comparison results and explaining the architecture choice would be interesting.

On page 7 in the footnote, what is meant by the low spatial frequency of relevant forces, which the Digit is not optimized for? The Digit sensor provides a very high resolution spatial image, but no forces without calibration.

The manuscript references Section D in the Appendix for the Human subject study, but it should be Section C

Minor points:

Clarity could be improved by giving more assertive names to the sensor readings vs the additional external observation of the body I (e.g. the MRI scan). As reading and observation are often used interchangeably, this was confusing for me. Calling the MRI Scan ground truth or something similar could resolve this.

This is a very minor point, but the use of the word "slime" seems inappropriate. The Oxford Dictionary defines slime as "​any unpleasant thick liquid substance". As I do not think you want to convey judgment about whether the substance, representing human breasts, is pleasant or unpleasant to touch, the word "gel" or another substitute could be considered.

Comments about the answer to the Ethics question:
The paper includes a human subject study with 10 participants, but does not mention whether this study was approved by the IRB or any other ethics board. This should be clarified.

**Ethical Concerns:**

["Major Concern: Improper research involving human subjects"]

**Final Justification:**

After considering the authors’ thorough rebuttal and new experimental results, I am satisfied that my main concerns have been addressed. The authors have provided convincing additional evidence, clarified key methodological choices, and outlined clear plans for addressing outstanding issues. In light of these improvements and the overall contribution of the work, I have updated my score and recommend acceptance.

**Limitations:**

yes

**Quality:**

3

**Strengths And Weaknesses:**

Strengths:

- The paper tackles the challenging task of perceiving softness differences by tactile sensing, and is clearly structured and written (besides a few issues detailed below).
- While the representation learning method employs established encoder-decoder architectures with sequence models (GRU, MLPs, positional encoding), its application appears novel in the domain of tactile representation learning for soft body perception.
- A new way of manufacturing breast phantoms with different layers and embedded lumps of various sizes is introduced, enabling systematic experimentation.
- Extensive results on a custom FEM-based simulation are presented to support the main claim that the learned representations can capture intricate patterns in tactile measurements beyond a simple force map.

Weaknesses:
- It is questionable whether the promising results obtained in simulation transfer effectively to the real world, as in-depth experiments addressing this gap are missing.
- As a consequence, the significance and practical impact of the findings remain unclear.
- The machine learning architecture itself is not novel, so the empirical findings are more important

---

> ### Author Rebuttal · Authors · 2025-07-30
>
> We thank the reviewer for the insightful review.
>
>
> First, following a careful investigation of our data after the submission deadline, we found that our Xela sensor had a time-varying bias in the measurements. Fixing for this bias (by subtracting the first measurement in each trial) dramatically improved our real-world results (e.g., an F1 score improvement from 20% to 65% and real imaging results are very similar to simulated ones). We report these fixed results below, including additional experiments suggested by the reviewers.
> # Additional Results
> All experiments were repeated with 3 random seeds.
>
>
> ## Sensor Calibration (SC)
> Modified results based on subtracting the first measurement in each trial.
>
> | **Method**    | **Size Error [%]** | **CoM Error [mm]** | **F1 Score** |
> | -------- | ------- | ------- | ------- |
> | Average Prediction  |  51.9 ± 47.5 | 10.7 ± 3.6 | N/A |
> | MLP | 24.1 ± 0.2 | 7.6 ± 0.1 | N/A |
> | GRU | 9.5 ± 5.7 | 9.7 ± 1.3 | 19.5 ± 0.3 |
> | MLP + SC |  11.0 ± 1.1   | 3.7 ± 0.1 | N/A |
> | GRU + SC | 27.9 ± 1.8 | 3.1 ± 0.1 | 65.3 ± 0.2 |
>
> ## Learning Architecture
> As requested by multiple reviewers, we have performed an ablation study on the representation learning architecture, as well as the imaging predictor.
> We tested with a Transformer (TR) [1]  instead of a GRU and a Flow Matching (FM) [3] instead of the Transposed Convolution (TC).
> We also tried using a masked autoencoder (similar to [2]), where we mask 20% of the trajectories and reconstruct them using the decoder.
>
> | **Method**  | **Size Error [%]** | **CoM Error [mm]** |
> | -------- | ------- | ------- |
> | GRU + TC + SC | 27.9 ± 1.8 | 3.1 ± 0.1|
> | GRU + FM + SC | 20.857 ± 0.9 | 3.2 ± 0.1 |
> | TR + FM + SC | 35.4 ± 0.7  | 5.9 ± 0.3 |
> | Masked TR + FM + SC | 57.9 ± 2.0 | 8.7 ± 0.1 |
> | Masked GRU + FM + SC | 23.9 ± 1.0 | 3.5 ± 0.1 |
>
> The GRU, together with a Flow Matching predictor, works best. We believe that this experiment, together with the sensor calibration results, shows that the GRU is a reasonable choice.
>
> ## Force Map Learnable Baseline
> As suggested by the reviewer, we added a learnable predictor with the suggested force map as a representation.
>
> | **Method**    | **Size Error [%]** | **CoM Error [mm]** |
> | -------- | ------- | ------- |
> | Force Map | 262.9 ± 234.4 | 15.9 ± 13.0 |
> | Force Map + FM | 49.7 ± 2.4 | 5.1 ± 0.1 |
>
> We can see that our approach outperforms this baseline by a large margin, indicating the effectiveness of our self-supervised representation learning.
>
> ## Data Scaling
> As suggested by reviewer hsJh, to further support the claim of the scalability of our approach, we have collected more data (x1.7 from what was presented in the paper, collected after the submission deadline), and performed two additional experiments:
> 1. Pretrained the self-supervised encoder on the x1.7 data, while training the downstream tasks on the original (x1) data:
> | **Method** | **Size Error [%]** | **CoM Error [mm]** | **F1 Score** |
> | -------- | ------- | ------- | ------- |
> | MLP + SC |  11.0 ± 1.1   | 3.7 ± 0.1 | N/A |
> | GRU + FM + SC | 20.857 ± 0.9 | 3.2 ± 0.1 | 65.3 ± 0.1 |
> | GRU + FM + SC x1.7 rep data | 18.6 ± 1.5 | 2.4 ± 0.0 | 72.7±0.3 |
> | MLP + SC x1.7 rep data | 11.7 ± 0.4 | 3.1±0.1 | N/A |
> 2. Pretrained both the self-supervised encoder and downstream tasks on the x1.7 data (as well as x0.5 and x0.25 data, as shown in the paper). The results are as follows:
> | **Method** | **Size Error [%]** | **CoM Error [mm]** | **F1 Score** |
> | -------- | ------- | ------- | ------- |
> | GRU + FM + SC x0.25 data | 99.8 ± 0.1 | 8.7 ± 0.2 | 0.8 ± 0.4 |
> | GRU + FM + SC x0.5 data | 41.6 ± 3.2 | 5.4 ± 0.4 | 43.9 ± 2.6 |
> | MLP + SC x1.7 data | 10.5 ± 0.5 | 2.6 ± 0.1 | N/A |
> | GRU + FM + SC x1.7 data| 23.0 ± 2.1 | 2.1±0.0 | 74.4±0.1 |
> | Force Map + FM x1.7 data | 46.6 ± 2.8 | 4.7 ± 0.1 | 47.3 ± 1.3 |
>
> Observe that additional data helps more in the self-supervised pretraining phase than to downstream tasks. This further validates our self-supervised representation learning approach.
>
> # Simulation Design
> We emphasize: the simulation is mainly for quick iteration on design choices in the learning algorithm and data collection setup. Since the simulation is 2D and quite simplistic, *it is not meant for sim2real transfer*, and we do not expect the results in the real world to be the same as those in the simulation.
> Our real world data collection and training required significant time and resources, and it was important to establish general guidelines from simulation beforehand: the number of different objects required (hundreds), the length of palpation trajectories (tens of samples), the number of trajectories per model (tens), the factors of variation in the objects (lump size and location). In the future, we believe the simulator will be useful for (1) investigating different representation learning architectures with much larger data, (2) investigating active learning for data collection, among other ideas.
> Insights that we validated that transfer from simulation:  (1) Other than the sensor force measurement and location dimensions, our self-supervised learning and downstream predictors are the same; (2) We show in the new Data Scaling results above that the simulation scaling law, presented in Figure 3d, is also valid in the real world.
> Therefore, we see the simulation as one of the contributions of the paper. We did not perform an ablation study on the number of sensors on the probe since we cannot change this in our hardware. For the “poke” trajectory's angle, we believe this is a minor detail; an improved data collection policy that we will pursue in future work would rotate the probe in various angles like a human. If the reviewer insists, we can also incorporate this study in the revised version.
> # “A better baseline in my opinion would be to compare the learning-based method on the force maps as a representation”:
> We thank the reviewer for the excellent idea.
> The results of a learning-based method on the force maps as a representation can be seen in the additional results above. Note that the advantage of self-supervised learning is evident both from these results but also from Figure 3d - we improve the representation just from self-supervised interactions, without image labels. This is not possible in a non-learned representation like the force maps.
> # “In the method formulation, a tactile sensor is described as a rigid body”
> For all practical purposes, both the Xela sensor and a common vision-based tactile sensor can be assumed to be rigid, as they are several orders of magnitude harder than the phantoms in our experiments. This can clearly be seen for the Xela sensor in the videos on the supplementary project website.
> To further support this, we measured the deformation of different sensors under 5N force (the largest force applied during our measurements). The Xela sensor was deformed by 0.19+-0.04 [mm], and a Digit vision-based sensor [4] was deformed by 0.18+-0.02 [mm].
> We are not aware of significantly softer tactile sensor products. This may be an interesting future direction to explore.
> # Representation Learning Architecture
> GRUs (or any RNNs for that matter) memory consumption depends linearly in the sequence length, compared to transformers, which scale quadratically. Since in our case the sequences are quite long (up to 50,000 sensor readings per sequence), this became a significant issue and required significant computing resources to run.
> We ran an experiment showing that GRU also outperformed a vanilla Transformer in this setting (and other architectures), which can be seen in the results above. We acknowledge that there is still plenty to experiment with in this direction, but hypothesise that the data is more important than the representation learning architecture in this setting (as evident from our sensor calibration and data scaling results above).
> # “On page 7 in the footnote, what is meant by the low spatial frequency of relevant forces, which the Digit is not optimized for? The Digit sensor provides a very high resolution spatial image, but no forces without calibration.”
> The digit is excellent at capturing high frequency variations in the image, for example, when pressing on a coin or cloth, the digit can identify the small changes in pressure that the coin indentation or cloth fibers apply. However, when pressing inside the soft phantom, a relatively similar pressure is applied on all the image pixels, and the sensor is effectively saturated. See Figure 6d for an example. Nevertheless, extending our approach to use gelsight sensors in some way is an exciting direction for future study.
> # Human-Study Ethical Concern
> The experiment required each participant to touch something like a squishy toy for several minutes and answer a question. There was absolutely no danger or discomfort involved.
> We have now obtained IRB approval. Since our institution does not allow retroactive approval, we will redo the experiment (under exactly the same conditions).
> In any case, this preliminary study was meant to give an impression of the difficulty of the task. A more extensive evaluation will be required to fully calibrate the performance of this technology, which is deferred to future work.
>
>
>
>
> # Clarity and Writing Comments
> We thank the reviewer for carefully reading our paper and finding these issues.
> We will make the following changes in the revised version:
> 1. Fix the references for the Human subject study in the Appendix.
> 2. Change the notation used for sensor reading and MRI scans (will indeed call the MRI scans ground truth as suggested).
> 3. Change “slime” to “gel” throughout the text.
>
> [1] Vaswani et al. Attention Is All You Need, 2017
>
> [2] He et al. Masked Autoencoders Are Scalable Vision Learners, 2021
>
> [3] Lipman et al. Flow Matching for Generative Modeling, 2022
>
> [4] Lambeta el al. DIGIT: A Novel Design for a Low-Cost Compact High-Resolution Tactile Sensor with Application to In-Hand Manipulation, 2020

---

> > ### Author Response · Authors · 2025-08-07
> >
> > Dear reviewer, we thank you again for your insightful review.
> >
> > We believe to have answered your concerns in our rebuttal and would be happy if you could take a look and see if it changes your score, or let us know if you have any further questions.

---

> > > ### Author Response · Authors · 2025-08-08
> > >
> > > Dear reviewer,
> > >
> > > There are less than 24 hours left for the discussion period.
> > >
> > > We would be happy to discuss any further concerns you might have following our rebuttal.

---

### Official Review · Reviewer_hsJh · 2025-07-03

**Clarity:** 3
**Significance:** 3
**Originality:** 4
**Rating:** 4
**Confidence:** 3

**Summary:**

This paper proposes a self-supervised learning framework for artificial palpation using tactile sensors on soft objects. The authors design a simulation and collect real-world tactile data with MRI as ground truth. A representation is learned via an encoder-decoder, and its usefulness is demonstrated on imaging and change detection. The motivation and method are well explained, and the visual materials help understanding. However, the downstream tasks are limited, and the model is only tested at small scale, raising questions about real-world applicability.

**Questions:**

- How can you conclude pretraining helps at scale when only small datasets (x1, x0.5, x0.25) are used?

- Do you plan to add more downstream tasks to validate the learned representations?

**Ethical Concerns:**

["NO or VERY MINOR ethics concerns only"]

**Final Justification:**

The rebuttal strengthens the paper with sensor calibration fixes, architectural ablations, a new force map baseline, and an additional downstream task. These address some of my earlier concerns, though I remain cautious about giving full weight to extensive post-deadline additions.

The dataset scale is now clear and impressive for this domain. However, the motivation for self-supervised learning could be better articulated—beyond the general observation that more data improves performance, it’s not fully clear why this approach is particularly valuable here.

The authors’ clarification on the long-term clinical goal is reasonable, and I suggest making this future plan more explicit in the paper. I maintain my borderline accept score.

**Limitations:**

- The experiments are done on a small scale and don’t show generalization to real clinical settings.

- The paper presents limited use cases for the learned representation.

**Paper Formatting Concerns:**

- Figure 5(c) needs more explanation in the caption.

- No major formatting issues otherwise.

**Quality:**

3

**Strengths And Weaknesses:**

Strengths:

- Very well-written introduction and motivation, especially in the medical context.

- The overall pipeline is clear and follows standard representation learning design.

- Simulator visualizations are helpful.

- Experiments show that the model performs better than force-map baselines.

Weaknesses:

- Only two downstream tasks are included, which is limited for a representation learning paper.

- The scale of the data is small. It's unclear how the method would generalize to real human data.

- Some results (e.g., pretraining on small-scale data) are stated strongly without strong support.

- Figure 5(c) is hard to understand and needs a better caption.

---

> ### Author Rebuttal · Authors · 2025-07-30
>
> We thank the reviewer for the insightful review.
>
>
> First, following a careful investigation of our data after the submission deadline, we found that our Xela sensor had a time-varying bias in the measurements. Fixing for this bias (by subtracting the first measurement in each trial) dramatically improved our real-world results (e.g., an F1 score improvement from 20% to 65% and real imaging results are very similar to simulated ones). We report these fixed results below, including additional experiments suggested by the reviewers.
>
> Rebuttal instructions forbid us from adding images to the anonymous URL in the paper. If the AC agrees, we will update the images on the website - our predictions are now much better, almost as good as in the simulation.
> # Additional Results
> All experiments were repeated with 3 random seeds.
>
>
> ## Sensor Calibration (SC)
> Modified results based on subtracting the first measurement in each trial.
>
> | **Method**    | **Size Error [%]** | **CoM Error [mm]** | **F1 Score** |
> | :------- | :-------: | :-------: | :-------: |
> | Average Prediction  |  51.9 ± 47.5 | 10.7 ± 3.6 | N/A |
> | MLP | 24.1 ± 0.2 | 7.6 ± 0.1 | N/A |
> | GRU | 9.5 ± 5.7 | 9.7 ± 1.3 | 19.5 ± 0.3 |
> | MLP + SC |  11.0 ± 1.1   | 3.7 ± 0.1 | N/A |
> | GRU + SC | 27.9 ± 1.8 | 3.1 ± 0.1 | 65.3 ± 0.2 |
>
> ## Learning Architecture
> As requested by multiple reviewers, we have performed an ablation study on the representation learning architecture, as well as the imaging predictor.
> We tested with a Transformer (TR) [1]  instead of a GRU and a Flow Matching (FM) [3] instead of the Transposed Convolution (TC).
> We also tried using a masked autoencoder (similar to [2]), where we mask 20% of the trajectories and reconstruct them using the decoder.
>
> | **Method**  | **Size Error [%]** | **CoM Error [mm]** |
> | :------- | :-------: | :-------: |
> | GRU + TC + SC | 27.9 ± 1.8 | 3.1 ± 0.1|
> | GRU + FM + SC | 20.857 ± 0.9 | 3.2 ± 0.1 |
> | TR + FM + SC | 35.4 ± 0.7  | 5.9 ± 0.3 |
> | Masked TR + FM + SC | 57.9 ± 2.0 | 8.7 ± 0.1 |
> | Masked GRU + FM + SC | 23.9 ± 1.0 | 3.5 ± 0.1 |
>
> The GRU, together with a Flow Matching predictor, works best. We believe that this experiment, together with the sensor calibration results, shows that the GRU is a reasonable choice.
>
> ## Force Map Learnable Baseline
> As suggested by reviewer SEnm, we added a learnable predictor with the suggested force map as a representation.
>
> | **Method**    | **Size Error [%]** | **CoM Error [mm]** |
> | :------- | :-------: | :-------: |
> | Force Map | 262.9 ± 234.4 | 15.9 ± 13.0 |
> | Force Map + FM | 49.7 ± 2.4 | 5.1 ± 0.1 |
>
> We can see that our approach outperforms this baseline by a large margin, indicating the effectiveness of our self-supervised representation learning.
>
> ## Data Scaling
>
>
> To further support our scalability claim, we have collected more data (x1.7 from what was presented in the paper, collected after the submission deadline), and performed two additional experiments:
> 1. Pretrained the self-supervised encoder on the x1.7 data, while training the downstream tasks on the original (x1) data:
> | **Method** | **Size Error [%]** | **CoM Error [mm]** | **F1 Score** |
> | :------- | :-------: | :-------: | :-------: |
> | MLP + SC |  11.0 ± 1.1   | 3.7 ± 0.1 | N/A |
> | GRU + FM + SC | 20.857 ± 0.9 | 3.2 ± 0.1 | 65.3 ± 0.1 |
> | GRU + FM + SC x1.7 rep data | 18.6 ± 1.5 | 2.4 ± 0.0 | 72.7±0.3 |
> | MLP + SC x1.7 rep data | 11.7 ± 0.4 | 3.1±0.1 | N/A |
> 2. Pretrained both the self-supervised encoder and downstream tasks on the x1.7 data (as well as x0.5 and x0.25 data, as shown in the paper). The results are as follows:
> | **Method** | **Size Error [%]** | **CoM Error [mm]** | **F1 Score** |
> | :------- | :-------: | :-------: | :-------: |
> | GRU + FM + SC x0.25 data | 99.8 ± 0.1 | 8.7 ± 0.2 | 0.8 ± 0.4 |
> | GRU + FM + SC x0.5 data | 41.6 ± 3.2 | 5.4 ± 0.4 | 43.9 ± 2.6 |
> | MLP + SC x1.7 data | 10.5 ± 0.5 | 2.6 ± 0.1 | N/A |
> | GRU + FM + SC x1.7 data| 23.0 ± 2.1 | 2.1±0.0 | 74.4±0.1 |
> | Force Map + FM x1.7 data | 46.6 ± 2.8 | 4.7 ± 0.1 | 47.3 ± 1.3 |
>
> Observe that additional data helps more in the self-supervised pretraining phase than to downstream tasks. This further validates our claims.
>
> ## Additional Downstream Task
> As suggested by the reviewer, we added an additional downstream task: to predict the phantom shell that we used (4 different phantoms, each with slightly different gel mixture inside, and minor shape variations due to our manufacturing process). This information is not contained in the imaging task, as we only reconstructed the image of the insert, not of the outer phantom shell.
> We achieved a near-perfect classification accuracy of 99.6 ± 0.7, showing the representation contains effective information regarding the slight variations of phantom stiffness and shape (although all were produced in the same manner).
>
> Another potential task that our data supports is 3D reconstruction of the MRI scans (vs. our 2D slice results).
>
> If the reviewer has any further ideas for possible downstream tasks, we would be happy to incorporate them as well.
>
> # “The scale of the data is small”
> The reviewer claims, “The scale of the data is small”. We are not sure what the reviewer means - small compared to what exactly? Our data is several orders of magnitude larger than previous datasets for soft objects that we are aware of (see Section 2 in the paper), and consists of ~550 different phantom-insert combinations with more than one hundred MRI scans for ground truth and 30 million tactile reading samples.
>
> Nevertheless, to further support our claims, we ran additional experiments on an even larger-scale dataset (collected after the submission deadline), which can be seen in the Data Scaling results above.
>
>
> # Generalization to real clinical setting
> We agree, this study does not validate the relevance of our new technology to real humans, but only takes a step in that direction (as discussed in depth in our Discussion Section).
> A study on humans requires several hundred patients who were tested positive for breast cancer, have not yet undergone surgery, have had an MRI scan recently done or planned, and are willing to undergo one or more artificial palpation exams.
> Such a study takes several years to complete, and involves many costs: financial, logistical, and emotional (to the patients). It is part of our research plan (we’re already talking with a large hospital to support it), but before we must ready our technology: how to scan using a human operator, how to improve the imaging results, what are the accuracies that we expect to measure, how to make the scanning convenient as possible, how to make anatomically accurate phantoms, etc. This submission is the first of many steps in this project, and we believe the current stage is interesting to the ML community to improve the technology. In any case, we believe that real human findings will be a better fit for a medical journal than NeurIPS.
>
> The contributions of the current study include:
> 1. An extensive dataset, phantom manufacturing process, and simulation environment that can be used to improve the learning algorithms;
> 2. Promising preliminary results on tactile imaging and change detection;
> 3. Specific capability measurements for this technology (CoM and size errors), which can be improved upon in future studies.
>
>
>
>
> # “How can you conclude pretraining helps at scale when only small datasets (x1, x0.5, x0.25) are used? “
> Since we didn’t have more data, we proved our scalability claim using two results:
> 1. Simulation result - In Figure 3d, we show that pretraining on a vast amount of self-supervised data helps in downstream tasks, especially when a little amount of supervised data is available.
> 2. Real data result - at the time of submission, we had collected data from ~550 phantom-insert combinations. We obviously couldn’t prove what would happen if more data were to be collected, but to hypothesize over it, we showed that if we were to train on x0.5 and x0.25 of the amount of data, we would get a serious deterioration in performance (as can be seen in Table 1). This demonstrates the trend of improvement vs. data size.
>
>
> To further support this claim, we have now collected more data (x1.7 from what was presented in the paper) and performed two additional experiments, demonstrating similar trends. The results can be seen in the Data Scaling results above.
>
>
> # “Figure 5(c) is hard to understand and needs a better caption”
> The force map generation procedure is depicted in Appendix G. We acknowledge that it is not sufficiently clear in the main text and will incorporate a dedicated subsection within the results section to describe all the used baselines in detail.
>
> [1] Vaswani et al. Attention Is All You Need, 2017
>
> [2] He et al. Masked Autoencoders Are Scalable Vision Learners, 2021
>
> [3] Lipman et al. Flow Matching for Generative Modeling, 2022

---

> > ### Author Response · Authors · 2025-08-07
> >
> > Dear reviewer, we thank you again for your insightful review.
> >
> > We believe to have answered your concerns in our rebuttal and would be happy if you could take a look and see if it changes your score, or let us know if you have any further questions.

---

> > ### Comment · Reviewer_hsJh · 2025-08-07
> >
> > Thank you for the detailed rebuttal and all the additional experiments.
> >
> > The sensor calibration fix and the follow-up experiments (architecture ablations, learnable force map baseline, and data scaling) do add substantial strength to the paper. That said, I’m a bit unsure how much weight should be placed on results generated after the submission deadline, even though I understand they offer useful insights.
> >
> > The newly added downstream task (phantom classification) is a helpful addition and shows that the learned representation captures more than just insert-level structure. However, I still find the motivation for using self-supervised learning underexplored. While the results show that more data improves performance, this is generally expected and not specific to self-supervised approaches. **What’s missing is a clear justification for why representation learning is especially suitable or necessary in this context. For example, does it enable better generalization across tasks or phantom types? Does it reduce the need for labeled data in future downstream tasks?** Clarifying this would strengthen the ML framing and help position the work more clearly within the broader representation learning literature.
> >
> > On the dataset size, I appreciate the clarification. I had underestimated the scale and effort behind it—30M samples and 550+ phantom-insert combinations is indeed substantial for this domain.
> >
> > Finally, I understand that clinical validation is out of scope for this work and appreciate that this is a first technical step in a longer research agenda. It may be helpful to mention the long-term plan and hospital collaboration more explicitly in the paper to set expectations.
> >
> > Thanks again for the thoughtful responses and the clear effort behind this work.

---

> > > ### Author Response · Authors · 2025-08-07
> > >
> > > ## "Does it reduce the need for labeled data in future downstream tasks?"
> > > We showed exactly this in two experiments:
> > > 1. *In Simulation:* In Figure 3d, we showed that using self-supervised pretraining on a large amount of data makes it possible to solve imaging (a relatively hard downstream task) with much less labeled data.
> > > 2. *In Real-Data:* Motivated by the original concern of the reviewer, we have added a similar experiment to the above in the rebuttal.  In the “Data Scaling” results, we have shown that adding more *unsupervised* data helps almost as much as adding more *labeled* data - an increase from 65.3 F1 score to 72.7 with additional unsupervised data vs 74.4 with supervised data.
> > >
> > > Based on these results, we hypothesize that self-supervised pretraining will play a crucial role in scaling this approach to real-world human data, where ground-truth labels are harder to obtain.
> > >
> > > Please let us know if you have any additional questions on this.
> > >
> > >
> > > ## Clinical Validation
> > >
> > > Thank you, we will add a discussion on this topic to the paper.

---

### Official Review · Reviewer_1RZ3 · 2025-07-15

**Clarity:** 2
**Significance:** 3
**Originality:** 3
**Rating:** 4
**Confidence:** 3

**Summary:**

The paper explores quantifying palpation forces in tissues with motivation for breast cancer imaging and the need to reduce subjectivity as well as detect/quantify subtle changes.
They do this via computer-based simulation of lumps within tissues as well as phantom-based experiments showing promising results in predicting MRI slices from aggregated reprsentation of sequence of palpation forces measured by the tactile sensors.
They show that self-supervised learning on sequences, (predicting future or past forces from a sequence), can be useful in representation learning and excels in low-data regime.
The self-supervised learning is modelled as non-autoregressive sequence-to-sequence prediction task.
Exploration of Recurrent Neural Network-based architecture (specifically, GRU) was done (although transformed-based architecture also resulted in similar results.)

**Questions:**

The results section could be made more clear.

Please explain why "In contrast with the simulated predictions, ..., We do not find the F1 score to be informative for interpreting image quality"?

In Table 1, the explanation of what these methods  "Image Pred", "Linear Pred.", "Force Map" mean is scattered throughout the paragraphs.
It would have been useful for reader to have these explanations done upfront in one-place including additional row for "naive baseline" that predicts the average lump area and CoM for all examples.

**Ethical Concerns:**

["NO or VERY MINOR ethics concerns only"]

**Limitations:**

yes

**Quality:**

2

**Strengths And Weaknesses:**

The paper is easy to follow (results section could be improved).

The paper explores quite novel and challenging task of tactile imaging via artificial palpation using force sensors to explore imaging the tumor structure within a tissue, albeit in a simplified setting of simulations and phantoms.
Given the rigor, extensiveness of the setup and the obtained data, the work is significant and original.

On the other hand, method-wise, limited exploration of the representation learning within the self-supervision framework has been done.
But given the dataset being made available publicly, the community can build upon the work and explore methodological innovations and improve the performance on the downstream clinical metric, in this case, extracting size and location of the lump, which remains far from clinically permissible.

---

> ### Author Rebuttal · Authors · 2025-07-30
>
> We thank the reviewer for the insightful review.
>
>
> First, following a careful investigation of our data after the submission deadline, we found that our Xela sensor had a time-varying bias in the measurements. Fixing for this bias (by subtracting the first measurement in each trial) dramatically improved our real-world results (e.g., an F1 score improvement from 20% to 65% and real imaging results are very similar to simulated ones). We report these fixed results below, including additional experiments suggested by the reviewers.
>
> Rebuttal instructions forbid us from adding images to the anonymous URL in the paper. If the AC agrees, we will update the images on the website - our predictions are now much better, almost as good as in the simulation.
> # Additional Results
> All experiments were repeated with 3 random seeds.
>
> ## F1 as a Palpation Imaging Metric
> We thank the reviewer for this question. We did not present the F1 score for two reasons:
> 1. Compared to the CoM and Size prediction errors, the F1 score is less intuitive.
> 2. In our real-world results, the F1 score was often zero (for a lump prediction without overlap with the ground truth).
>
>
> In our amended results, following the Xela bias fix, the F1 is informative, and we include it.
> ## Sensor Calibration (SC)
> Modified results based on subtracting the first measurement in each trial.
>
> | **Method**    | **Size Error [%]** | **CoM Error [mm]** | **F1 Score** |
> | :------- | :-------: | :-------: | :-------: |
> | Average Prediction  |  51.9 ± 47.5 | 10.7 ± 3.6 | N/A |
> | MLP | 24.1 ± 0.2 | 7.6 ± 0.1 | N/A |
> | GRU | 9.5 ± 5.7 | 9.7 ± 1.3 | 19.5 ± 0.3 |
> | MLP + SC |  11.0 ± 1.1   | 3.7 ± 0.1 | N/A |
> | GRU + SC | 27.9 ± 1.8 | 3.1 ± 0.1 | 65.3 ± 0.2 |
>
> ## Learning Architecture
> As requested by multiple reviewers, we have performed an ablation study on the representation learning architecture, as well as the imaging predictor.
> We tested with a Transformer (TR) [1]  instead of a GRU and a Flow Matching (FM) [3] instead of the Transposed Convolution (TC).
> We also tried using a masked autoencoder (similar to [2]), where we mask 20% of the trajectories and reconstruct them using the decoder.
>
> | **Method**  | **Size Error [%]** | **CoM Error [mm]** |
> | :-------- | :-------: | :-------: |
> | GRU + TC + SC | 27.9 ± 1.8 | 3.1 ± 0.1|
> | GRU + FM + SC | 20.857 ± 0.9 | 3.2 ± 0.1 |
> | TR + FM + SC | 35.4 ± 0.7  | 5.9 ± 0.3 |
> | Masked TR + FM + SC | 57.9 ± 2.0 | 8.7 ± 0.1 |
> | Masked GRU + FM + SC | 23.9 ± 1.0 | 3.5 ± 0.1 |
>
> The GRU, together with a Flow Matching predictor, works best. We believe that this experiment, together with the sensor calibration results, shows that the GRU is a reasonable choice.
>
> ## Force Map Learnable Baseline
> As suggested by reviewer SEnm, we added a learnable predictor with the suggested force map as a representation.
>
> | **Method**    | **Size Error [%]** | **CoM Error [mm]** |
> | :------- | :-------: | :-------: |
> | Force Map | 262.9 ± 234.4 | 15.9 ± 13.0 |
> | Force Map + FM | 49.7 ± 2.4 | 5.1 ± 0.1 |
>
> We can see that our approach outperforms this baseline by a large margin, indicating the effectiveness of our self-supervised representation learning.
>
> ## Data Scaling
> As suggested by reviewer hsJh, to further support the claim of the scalability of our approach, we have collected more data (x1.7 from what was presented in the paper, collected after the submission deadline), and performed two additional experiments:
> 1.  Pretrained the self-supervised encoder on the x1.7 data, while training the downstream tasks on the original (x1) data:
> | **Method** | **Size Error [%]** | **CoM Error [mm]** | **F1 Score** |
> | :------- | :-------:| :-------: | :-------: |
> | MLP + SC |  11.0 ± 1.1   | 3.7 ± 0.1 | N/A |
> | GRU + FM + SC | 20.857 ± 0.9 | 3.2 ± 0.1 | 65.3 ± 0.1 |
> | GRU + FM + SC x1.7 rep data | 18.6 ± 1.5 | 2.4 ± 0.0 | 72.7±0.3 |
> | MLP + SC x1.7 rep data | 11.7 ± 0.4 | 3.1±0.1 | N/A |
>
> 2. Pretrained both the self-supervised encoder and downstream tasks on the x1.7 data (as well as x0.5 and x0.25 data, as shown in the paper). The results are as follows:
> | **Method** | **Size Error [%]** | **CoM Error [mm]** | **F1 Score** |
> | :------- | :-------: | :-------: | :-------: |
> | GRU + FM + SC x0.25 data | 99.8 ± 0.1 | 8.7 ± 0.2 | 0.8 ± 0.4 |
> | GRU + FM + SC x0.5 data | 41.6 ± 3.2 | 5.4 ± 0.4 | 43.9 ± 2.6 |
> | MLP + SC x1.7 data | 10.5 ± 0.5 | 2.6 ± 0.1 | N/A |
> | GRU + FM + SC x1.7 data| 23.0 ± 2.1 | 2.1±0.0 | 74.4±0.1 |
> | Force Map + FM x1.7 data | 46.6 ± 2.8 | 4.7 ± 0.1 | 47.3 ± 1.3 |
>
> Observe that additional data helps more in the self-supervised pretraining phase than to downstream tasks. This further validates our approach.
>
>
> # Results Section Clarity
> We will add a sub-section explaining all of the used baselines in the revised version.
> We appreciate the suggestion of adding the naive, average prediction baseline; we agree it might help to ground the presented results. We added this baseline to our updated Table 1 (see above).
> If the reviewer has any further ideas on how to improve the Results Section, please let us know.
>
> # “...method-wise, limited exploration of the representation learning within the self-supervision framework has been done”
> As can be seen in the results above, we now include our results from experimenting with different representation learning architectures and imaging predictors.
>
>
> [1] Vaswani et al. Attention Is All You Need, 2017
>
> [2] He et al. Masked Autoencoders Are Scalable Vision Learners, 2021
>
> [3] Lipman et al. Flow Matching for Generative Modeling, 2022

---

> > ### Author Response · Authors · 2025-08-07
> >
> > Dear reviewer, we thank you again for your insightful review.
> >
> > We believe to have answered your concerns in our rebuttal and would be happy if you could take a look and see if it changes your score, or let us know if you have any further questions.

---

> > > ### Author Response · Authors · 2025-08-08
> > >
> > > Dear reviewer,
> > >
> > > There are less than 24 hours left for the discussion period.
> > >
> > > We would be happy to discuss any further concerns you might have following our rebuttal.

---

### Note · Authors · 2025-08-13

The reviewers found our domain novel and challenging (1RZ3, SEnm), our paper well written (hsJh, SEnm), and our dataset substantial, original, and may enable systematic experimentation (1RZ3, hsJh, SEnm). E.g., “Given the rigor, extensiveness of the setup and the obtained data, the work is significant and original.” (1RZ3)

Several concerns were raised – some addressed with new results, and others due to misunderstandings.

**There was almost zero engagement, despite several reminders**. Only hsJh addressed our rebuttal, raising another question, addressed by two of our results that the reviewer might have missed. That comment went unacknowledged. 1RZ3 and SEnm haven’t engaged at all. **We are convinced that a proper consideration of our rebuttal should increase the scores**.

## 1RZ3

Main concern: limited exploration of representation learning. Our rebuttal provides an extensive exploration of representation learning and downstream architectures. We believe our rebuttal addresses all of 1RZ3’s concerns.

## hsJh

Only remaining post-rebuttal concern: “a clear justification for why representation learning is especially suitable, …does it reduce the need for labeled data…?”. We addressed exactly this both in Figure 3d and in our rebuttal, showing that additional self-supervised learning significantly improves results without additional labeled data. The results, and our additional comparisons in answer to SEnm, clearly justify our approach.

## SEnm

Extensive review, but no post-rebuttal engagement.

1. Sim2real transfer - we believe there is a misunderstanding here. We explained this in the rebuttal, including new results showing real-world image predictions closely match the simulation results.

2. Compare with force maps as representation - great idea, we tried it, and our results clearly show the advantage of our learned representation.

3. Exploration of additional representation learning architectures - we did that quite extensively, including transformer architectures and flow matching for image predictions.

## Ethics Concerns

We addressed this (including IRB approval), but we can remove the experiment (it is only to convey the difficulty of the task, which the reviewers already acknowledged).

## Summary

**We didn’t get a fair chance at a discussion with the reviewers**, although we have answered most (if not all) of their concerns in our rebuttal. We ask the AC and reviewers to take this into account when finalizing their decisions.

---

### Decision · Program_Chairs · 2025-09-17

**Decision:**

Accept (poster)

**Comment:**

This paper describes a preliminary study designed to investigate the feasibility of using palpation data collected with a force sensor on a robotic arm to build a representation of a soft object that could be used to provided clinically relevant data such as the size and location of possible tumors or changes in a embedded mass over time that could be indicative of tumor growth.

The paper investigates the efficacy of the representations built from the collected data using a neural representation by comparing predictions made with the model to MRI images which are used as ground truth. It also presents results on simulated data evaluating the utility of building a representation model vs building a model solely based on the data acquired from a particular simulated phantom.

Some of the papers strengths include:

A reasonably detailed 2D simulation experiment which provides some initial evidence for the approach under controlled settings where ground truth is known.

A fairly extensive data set collected on a custom designed phantom where the position of the inserted tumor was varied and the force sensor and MRI data are collected.

Preliminary results which support the claims made in the paper regarding the possible utility of this palpation modality for tumor sensing.

Some weaknesses include:

The lack of data on human subjects since that was beyond the scope of this preliminary study.

Concerns regarding IRB approval for a small human subject experiment which was designed to provide a baseline of a humans ability to make decisions based on palpation.


As a preliminary study this paper provides some initial evidence about the utility of exploring palpation data for anatomical analysis. The representation learning techniques applied are fairly standard but the domain is novel and the authors present arguments for the design choices they have made.

The authors provided a detailed rebuttal that addressed most of the reviewers concerns about data set size and design decisions for their approach to representation learning. They also provided a path to address concerns related to the use of human subjects by stating that the experiments could be re-run with IRB approval and by addressing in the document the questions raised by the ethics reviewers.